

# Warmer growing seasons improve cereal yields in Northern Europe only with increasing precipitation

Faranak Tootoonchi[1,2*], Göran Bergkvist[1], Giulia Vico[3]

1 Department of crop production ecology, Swedish University of Agricultural Sciences (SLU), Uppsala, Sweden

5   2 Department of Earth and Atmospheric Sciences, Cornell University, Ithaca, NY, USA

Department of ecology, Swedish University of Agricultural Sciences (SLU), Uppsala, Sweden

[*] *Correspondence to*: Faranak Tootoonchi (faranak.tootoonchi@slu.se)

**Abstract.** Crop yields depend on climatic conditions such as precipitation and temperature and their timing before and during the growing season. At high latitudes, climate change could lengthen the growing season and provide

more suitable temperatures, but also expose crops to more frequent damaging conditions. We quantified the response of regionally-averaged 1965-2020 winter and spring cereal yields in Sweden to a wide set of descriptors of climatic conditions. With statistical models, we explored the role of both short-term and average conditions over physiologically relevant developmental stages, as well as of a proxy of water availability during the period prior to the main growing season. Temperature and precipitation or dry spell lengths for the entire growing season

explained 75-85% of yield variability, performing better than short-term potentially damaging conditions. Low precipitation or extended dry spells combined with high temperatures and, conversely, high precipitation sums with cool temperatures decreased yields for all crops. Our findings suggest that under climate change crop yields will be reduced in Sweden, unless warming is accompanied by increase in precipitation during the main growing season. With unaltered or reduced growing season precipitation, benefiting from warmer temperatures caused by

climate change will require adaptation measures.

**Keywords.** Climate change, precipitation and temperature, climatic indicators, crop yield, cereals, high latitudes



## 1 Introduction

Crop yields depend on climatic conditions such as precipitation, temperature, and their interactions (Porter and Semenov, 2005; Ray et al., 2015; Riha et al., 1996). These climatic conditions define the water and energy available for crop establishment, growth, and yield formation (Barron-Gafford et al., 2012; Song et al., 2016). Crop responses to climatic conditions are complex and nonlinear. Both excessive and insufficient precipitation, as well as excessively high or low temperatures, can damage the crop and reduce marketable yields (Hatfield and
Prueger, 2015; Luan et al., 2021; Miedema, 1982; Mittra and Stickler, 1961). Excessively dry and warm conditions can cause water and heat stress, hasten development, reduce net photosynthesis, kernel numbers and size, and ultimately decrease crop yield (Hatfield and Prueger, 2015; Praba et al., 2009; Siebert et al., 2017). At the other extreme, excessive precipitation is often associated with reduced solar radiation (Díaz-Torres et al., 2017), causes nutrient losses and oxygen deficiency (Becker, 2014; Schreiber, 1999), and enhances fungal growth (Barnes et al.,
2018). Intense precipitation can also damage crops mechanically. Identifying the most damaging conditions for crop yields and determining the extent of their impacts are necessary steps to assess expected climate change effects on crop yields and for improving crop yield modeling.

At high latitudes, such as in Northern Europe, current temperatures lead to short periods for soil cultivation and crop growth. Global warming can create new opportunities, by contributing to longer growing season and
more suitable growing conditions, potentially increasing yields (Bindi and Olesen, 2011; Juhola et al., 2017; Olesen et al., 2011; Wiréhn et al., 2017) or widening the range of cultivable crops (Heikonen et al., 2025; Wiréhn, 2018). At the same time, warmer conditions enhance evapotranspiration thus reducing soil water availability and increasing the risk of plant water stress unless precipitation increases. Moreover, frequency and magnitude of extreme conditions such as dry, hot or wet spells, are expected to increase globally, including in northern Europe
(Song et al., 2016; Toreti et al., 2019; Wiréhn et al., 2017). Indeed, reduced yields due to excessively dry periods during the growing season or due to soil water saturation that limits field access to machinery, are already observed in Northern Europe (De Toro et al., 2015; Trnka et al., 2011). Nonetheless, the net effect of positive and negative changes of climatic conditions in Northern Europe is still unclear.

Different aspects of climatic conditions and their co-occurrence affect crop yields. Even detrimental conditions
of short duration can have large adverse effects (Hakala et al., 2020; Zhu and Troy, 2018). Averaging the climatic conditions over the growing season can mask the role of these short-term, but potentially severely damaging conditions (Lesk et al., 2022; Luan et al., 2021; Troy et al., 2015; Vogel et al., 2019). Considering short-term conditions is particularly important in the face of climate change because of the projected changes in timing and magnitude of short-term extreme events (Thiery et al., 2021; Tootoonchi et al., 2023; Wiréhn et al., 2017).
Furthermore, combinations of damaging climatic conditions can have impacts on crops that are disproportionally large compared with the sum of their individual impacts (Alizadeh et al., 2020; Luan et al., 2021; Suzuki et al., 2014; Teutschbein et al., 2023a). To effectively evaluate the effects of climatic conditions on crop yields, we need to consider both seasonal average and short-term conditions, as well as their combinations. Yet we lack quantification of the role of short-term conditions and combination of conditions on crop yields at high latitudes.

Crops respond differently to growing conditions at different developmental stages, with often complex and non-linear mechanisms (Lüttger and Feike, 2018; Mäkinen et al., 2018; Suliman et al., 2024; Trnka et al., 2014). For example, spring cereals tend to be more sensitive to water stress around flowering (Martyniak, 2008) compared





with other developmental stages, and even more so when co-occurring with heat stress (Barnabás et al., 2008; Dolferus et al., 2011; Senapati et al., 2021). Late-season precipitation can reduce crop yield and quality by

enhancing fungal growth or delaying harvests due to wet soils (Olesen et al., 2011). The conditions before the beginning of the growing season can also have legacy impacts, by affecting soil water accumulation (Trnka et al., 2016). Accumulated soil water before the beginning of the growing season helps buffering the negative effects of limited precipitation after sowing of spring crops or after the end of winter dormancy for winter crops (Li et al., 2019). Excessive soil water, however, delays sowing of spring crops (Trnka et al., 2011). Despite the importance

of the timing in relation to crop physiological development, the response to climatic conditions at different developmental stages, and the role of legacy effects of climatic conditions before the main growing season, remain largely uncharacterized, in particular at high latitudes.

Climatic conditions during a period can be summarized via a variety of climatic indicators (McMillan, 2021; Schmidt and Felsche, 2024; Wiréhn, 2021; Zhu and Troy, 2018). These are 'simple diagnostic quantities that are

used to characterize an aspect of a geophysical system' (Schneider et al., 2013), such as states, variability, or frequencies of climatic conditions (Wiréhn, 2021). Selecting the most suitable indicators is challenging and depends on the purpose, regional climate and most critical climatic conditions (Adger, 2006; Wiréhn, 2021). A systematic exploration of a wide range of relevant climatic indicators is essential to identify the most relevant indicators to explain crop yields.

Using county-level yield data of staple cereals and meteorological data for 1965-2020 across Sweden, we systematically explore the role of various climatic indicators on crop yields under Northern European conditions. Specifically, we answer the following questions:

1. Which climatic conditions and relative to which period (i.e., pre- (main) growing season, pre- or post-flowering, or whole growing season) explain crop yields best?

2. Which climatic conditions, and their combinations, are most beneficial for yields? Which are most detrimental?

The results can help predict crop yield through statistical models. They also provide insights into the climate change impacts on agriculture for Northern Europe and whether specific cereals are better adapted to the conditions expected to become more frequent in the future.

## 2  Data and Methods

### 2.1  Crop yield and climatic data

Crop yield data for the period 1965-2020 over the 21 Swedish counties (län in Swedish) was obtained from Statistics Sweden (Statistikdatabasen), for the most commonly grown cereals in the country, i.e. winter wheat, spring wheat, spring barley and oats (Fig. 1). The percentage of unavailable data was 18%, 33%, 1% and 13%,

respectively, and can be ascribed to either uncertainty in yearly yield cultivation (i.e., yield not meeting reporting criteria) or confidentiality reasons (i.e., crop sown only in few fields, Statistikdatabasen). The fraction of cultivated area per county was higher in the south where conditions are more favorable, permitting cultivation of all four crops, whereas only oats and spring barley were reported for the north (Fig. 1).



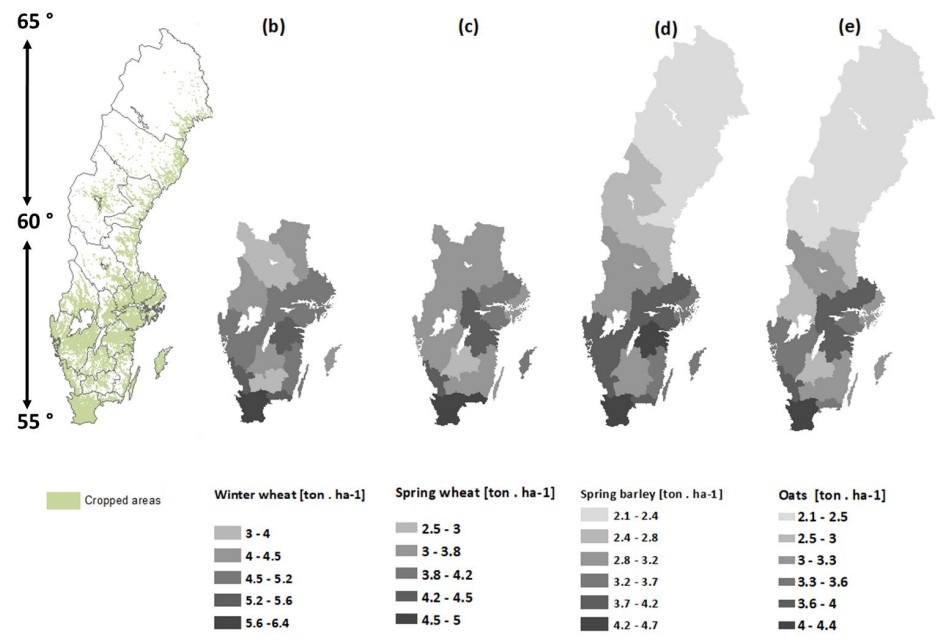

**Figure 1: Overview of the study area including (a) the location of cropped areas in green and (b-e) the average yields of each of the four crops for each county in Sweden during the period 1965-2020. Note that winter wheat and spring wheat are cultivated only in the southern most counties.**

For meteorological data, we used gridded observed precipitation as well as average, minimum and maximum daily temperature values from E-Obs gridded data (Cornes et al., 2018) version 26.0e at 0.1 ° resolution. Counties extend over 2.9-98.2 km² and include large areas that are not cropped due to unsuitable climatic and soil conditions. Hence, we considered as representative of each county the gridded meteorological data averaged over cropped areas of the county. The cropped area was assumed to match the the Non-irrigated arable fields in the CORINE land cover map (European Environment Agency, 2020). The 2006 map (CLC2006) was used as representative of the entire period.

## 2.2   Modeling components

We aimed to identify which climatic conditions, their combinations and timing during the year were most important in defining each crop yield. The climatic indicators were chosen based on our ecophysiological understanding of plant response to climatic conditions (as detailed in Section 2.2.1). We considered four periods of physiological relevance, defined for each crop and year based on a minimalist phenological model (Section 2.2.2). We then compared the performance of several statistical models, with yield of each crop as dependent variable and different climatic indicators and their interactions as explanatory variables (Section 2.2.3).



### 2.2.1 Selection procedure of climatic indicators

We selected candidate climatic indicators based on previous local and global analyses, to include those that
had shown promise to explain crop yield variability across a wide geographic ranges (Kaseva et al., 2023; Luan et
al., 2021; Mäkinen et al., 2018; Wiréhn et al., 2017; Zhu and Troy, 2018). We also considered some
complementary indicators that had not been previously used to explain yield variability, but we deemed relevant
in particular under Northern European conditions, based on the climatic conditions under which most commonly
grown crops thrive or are damaged.

The selection process resulted in 20 candidate indicators (Table 1). These either reflected attributes of
precipitation and temperature (averages over specific periods or frequency of exceedance of specific thresholds of
either precipitation or temperature), or were composite in nature (i.e., included the role of both temperature and
precipitation). Examples of composite indicators are the dryness index (DI) and total precipitation occurring in
days with freezing temperature (P_T0). Different attributes of the climatic conditions were captured, such as
variability (e.g., standard deviation of precipitation and temperature, Pvar and Tvar), duration of specific
conditions (e.g., maximum number of consecutive dry days, CDD), occurrence of peaks over thresholds (e.g.,
number of days with precipitation above 10 mm, NDP10) and magnitude of a variable (e.g., maximum
precipitation amount in a single day, MaxP_1D, or total precipitation over the period, Psum). The candidate
indicators also reflected conditions over different time spans, such as averages over longer periods, up to the whole
main growing season (e.g., mean temperature, Tmean), or short-term but potentially severely detrimental
conditions lasting only one or few days (e.g., CDD).

We checked the candidate indicators for similarity, based on Spearman correlation coefficients (Spearman,
1904, Fig. 2). To focus on complementary indicators and reduce redundancy and collinearity, we eliminated one
climatic indicator within each pair of highly correlated (>|0.5|) indicators (as in e.g. Addor et al., 2018 or Sjulgård
et al., 2023). The pairwise correlations between precipitation and temperature indicators were generally lower
(between -0.3 and 0.3) compared with correlation between indicators based on precipitation or temperature only.
A general positive correlation prevailed among precipitation indicators, except for indicators of dryness (Ndry5
and CDD), which were negatively correlated with the rest of the precipitation indicators. Temperature indicators
were negatively correlated for those representing low temperatures Frost and Icing days, as well as for daily
temperature variance, but were postively correlated for the rest of the indicators.

Among pairs of highly correlated indicators linked to precipitation, we chose CDD over NDDry_5d because
the former represents dry periods of various lengths. Between CWD and NDP1, we chose NDP1 as it may
indirectly capture also the reduction in solar radiation (Díaz-Torres et al., 2017). We also kept the number of days
with precipitation exceeding 20 mm (NDP20), instead of the indicator relative to 10 mm (NDP10), due to the
damaging and extreme nature of the former. We also eliminated the maximum amount of precipitation over 1 and
5 days (MaxP_1D and MaxP_5D) because both were highly correlated with NDP20. For highly correlated
temperature indicators, we selected the number of days above 25 °C (NDT25) instead of 30 °C (NDT30) because
temperatures higher than 25 °C likely are at the high end of optimal temperatures of typical local varieties;
moreover, the historically low frequency of days above 30 °C could reduce the robustness of the constructed
statistical models. Between number of frost and icing days we retained frost days, because freezing nights can be
sufficient to cause permanent damage in non-acclimated crops (François and Vrac, 2023). Regarding average



temperatures, we focused on Tmean, i.e., the average mean daily temperature, because it was well correlated with both minimum and maximum daily temperatures, averaged over the same period. Moreover, Tmean is readily available in climatic data. Between the two composite indicators reflecting DI and P_T0 showed high correlations, we selected DI, which is a proxy of soil water availability, because of the direct impact of soil moisture on crops (Jackson et al., 2023). Furthermore, we did not expect any legacy effect of the total precipitation in freezing conditions, P_T0, during pre-(main) growing, particularly on spring-sown crops, because those are cultivated after winter.

After the selection, we retained five precipitation indicators (Psum, Pvar, NDP1, NDP20 and CDD), four temperature indicators (Tmean, Tvar, frost and NDT25), and one composite indicator (DI; in bold in Table 1).

**Table 1: List of 20 indicators that were identified from the literature or were deemed relevant to assess the impact of climatic conditions on crop yields at high latitudes. Indicators were categorized into 3 categories, representing (a) precipitation (b) temperature and (c) combined influence of precipitation and temperature. The indicators in bold are those selected after a second round of screening, eliminating those showing high Spearman correlation with the selected ones, and after selecting those with complementary nature.**

| Variable | climatic indicators [Unit] | Description | Examples of application in crop response |
|---|---|---|---|
| (a)Precipitation | **Psum [cm]** | Total of daily precipitation | (Luan et al., 2022; Lüttger and Feike, 2018; Vogel et al., 2019) |
| | **NDP1[days]** | Number of days, also non-consecutive, with daily precipitation above 1 mm | (Copernicus Climate Change Service, 2019) |
| | NDP10 [days] | Number of days, also non-consecutive, with daily precipitation above 10 mm | (Copernicus Climate Change Service, 2019) |
| | **NDP20 [days]** | Number of days, also non-consecutive, with daily precipitation above 20 mm | (Copernicus Climate Change Service, 2019) |
| | MaxP_1D [mm·day$^{-1}$] | Maximum precipitation amount in one day | (Lesk et al., 2020) |
| | MaxP_5D [mm·day$^{-1}$] | Maximum precipitation amount in five consecutive days | (Vogel et al., 2019; Zhu and Troy, 2018) |
| | NDDry_5d [-] | Number of dry spells longer than 5 consecutive days | (Manning et al., 2023; Masupha et al., 2016) |
| | **Pvar [mm$^2$·day$^{-2}$]** | Variance of daily precipitation | (Tootoonchi et al., 2022; Zhu and Troy, 2018) |
| | **CDD [days]** | Maximum number of *consecutive* dry (< 1mm/day) days | (Luan et al., 2022) |
| | CWD [days] | Maximum number of consecutive wet (> 1mm/day) days | (Copernicus Climate Change Service, 2019) |
| (b)Temperature | **Tmean [°C]** | Mean of daily mean temperature | (Carter et al., 2018; Luan et al., 2022; Vogel et al., 2019) |
| | Tmin [°C] | Mean of daily minimum temperature | (Copernicus Climate Change Service, 2019) |
| | Tmax [°C] | Mean of daily maximum temperature | (Copernicus Climate Change Service, 2019) |
| | **NDT25[days]** | Number of days, also non-consecutive, with maximum temperature above 25°C | (Lüttger and Feike, 2018; Zhu and Troy, 2018) |
| | NDT30 [days] | Number of days, also non-consecutive, with maximum temperature above 30°C | (Lüttger and Feike, 2018; Zhu and Troy, 2018) |
| | **Frost [days]** | Number of days, also non-consecutive, with daily minimum temperature below 0°C | (Copernicus Climate Change Service,2019) |
| | Icing [days] | Number of days, also non-consecutive, with daily maximum temperature below 0°C | (Zhu and Troy, 2018) |
| | **Tvar [°C$^2$·day$^{-2}$]** | Variance of daily mean temperature | (Tootoonchi et al., 2022; Zhu and Troy, 2018) |
| (c) Combined P and T | **DI[mm·mm$^{-1}$]** | Dryness index, i.e., the ratio of total potential evapotranspiration (PET) calculated via Hamon (Hamon, 1961) to total precipitation | (Luan et al., 2022; Todorovic et al., 2022) |
| | P_T0 [mm·day$^{-1}$] | Sum of precipitation occurring when mean daily temperature is below 0°C | (Climate indicator - Snow | SMHI, 2023; Fontrodona-Bach et al., 2023) |

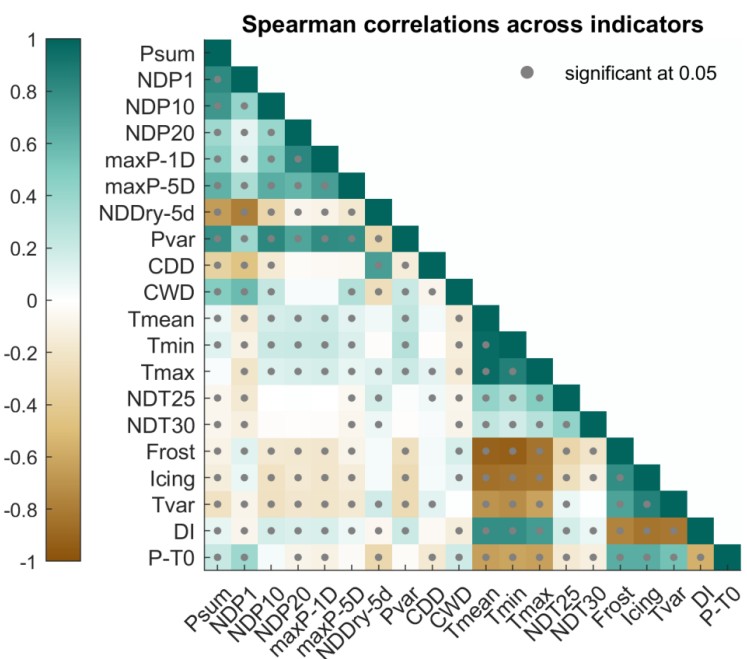

**Figure 2: Spearman correlation between pairwise indicators representing magnitude, variation and duration of precipitation and temperature, as well as two composite indicators over the entire growing season (see Table 1 for indicator definitions). The significant correlations (p < 0.05) are marked with a grey circle.**

### 2.2.2   Periods of interest

For each crop and cropped land in each county, we defined the (main) growing season, pre-flowering and post-flowering based on estimated sowing, flowering and maturity dates, averaged over the 56 years spanned by yield data. To estimate these dates, we used a phenological model based on growing degree days and day length, parameterized and already tested for crops across Europe (Marini et al., 2020; Olesen et al., 2012). For winter wheat the beginning of the main growing season was estimated as the first yearly occurrence of an increase in GDD corresponding steeper than $\geq 4$ °C·d$^{-1}$ (Costa et al., 2024). The date did not considerably change when other thresholds beyond 4 °C·d$^{-1}$ were considered. We defined the growing season as the period between sowing and maturity dates for spring cereals, and between beginning of the main growing season and maturity for winter wheat. Similarly, the pre-flowering period ran between sowing (for spring crops) and beginning of the main growing season (for winter wheat) and flowering, whereas the post-flowering period extends between flowering and maturity.

We calculated the retained indicators (Table 1, in bold) for each year and cropped land in each county, relative to the three crop-specific periods: entire growing season, pre- and post-flowering. Dryness index was calculated in the above-mentioned periods and also during pre-growing season periods. Lacking clear evidence on the duration of any legacy effects prior to sowing or beginning of the growing season, we considered the dryness index averaged over or 30, 60, 90 day-long periods before the sowing date for spring crops, or the beginning of the main growing period for winter wheat.



### 2.2.3    Linear mixed effect models

We used linear mixed effect models (Bürger et al., 2012; Smith et al., 2005) to explore crop yield responses to climatic indicators during the identified periods. We fitted separate models for each crop, with yield Y as the dependent variable, and a set of explanatory climatic indicators as fixed effects. Neither yields nor climatic indicators were de-trended. Instead, we included time elapsed from year 1965 ($t$) as a continuous variable in all models, thus accounting for the combined impacts of climate change and technological improvements, including

variety and management changes. Furthermore, year and county were included as categorical random effects in all models to consider spatiotemporal heterogeneity and variations over the study area.

We considered three types of models for each crop. The first type had the composite precipitation-temperature indicator, dryness index, as explanatory variable to capture the role of precipitation and temperatures at the same time. Dryness index can summarize key aspects of soil water availability essential for crop growth through

interplay between precipitation and temperature, thus be used as an indicator to capture balance between water supply and energy-driven demand. As such, dryness index provides a more process oriented view at the relationship between precipitation, temperature, and their interactions. We included also quadratic dependence of yield on dryness index to account for excessive dryness (Luan et al., 2022). The fixed part of the model was

$$Y = \beta_0 + \beta_t t + \beta_{DI}(DI) + \beta_{DI2}(DI)^2 \qquad \text{eq. 1}$$

Where $Y$ is the yield of the crop of interest, $\beta_0$ is the global intercept, $\beta_{\text{t}}$, $\beta_{DI}$ and $\beta_{DI2}$ represent dependences of

yield on t, DI and DI$^2$, respectively.

The second type of model aimed at quantifying the role of conditions that either extend over the whole period considered (e.g., averages), or for a potentially substantial part of that (e.g., dry spells). We focused on pairs of interacting indicators representing precipitation (x_P) and temperature (x_T) characteristics, separately. We considered three pairs of x_P and x_T indicators, each reflecting different aspects: i) Precipitation sums (Psum)

and temperature averages (Tmean), characterizing average conditions, ii) precipitation and temperature variance (Pvar and Tvar), characterizing the variability of the conditions, or iii) maximum length of the dry spells (CDD) and temperature averages (Tmean), characterizing the occurrence of dry spells, the effect of which is expected to be more marked at higher temperature. As fixed effects, in its most complex form, the model included a quadratic dependence on x_P and x_T as well as all the possible two-way interactions, because the damaging effects of low

precipitations are stronger with high temperatures and vice versa. The fixed part of the model was

$$\begin{aligned} Y = \beta_0 + \beta_t t + \beta_P(x\_P) + \beta_T(x\_T) + \beta_{P2}(x\_P)^2 + \beta_{T2}(x\_T)^2 + \beta_{PT}(x\_P)(x\_T) + \beta_{P2T}(x\_P)^2(x\_T) \\ + \beta_{PT2}(x\_P)(x\_T)^2 + \beta_{P2T2}(x\_P)^2(x\_T)^2 \end{aligned} \qquad \text{eq. 2}$$

where $\beta_0$ is the global intercept, $\beta_{\text{t}}$, $\beta_P$ and $\beta_T$ are the slopes of the linear dependencies of crop yield on time, precipitation and temperature indicators, respectively, and $\beta_{\text{P2}}$ and $\beta_{\text{T2}}$ are quadratic dependencies of crops to the same indicators. Including a quadratic dependence on x_P and X_T allows for intermediate yield-maximizing conditions. The interactions between precipitation and temperature indicators are represented by $\beta_{PT}$, $\beta_{P2T}$ and

$\beta_{PT2}$. We compared the performance of the most complex model (eq. 2) with nine model variants of decreasing complexity, for a total of 10 model variants for each of pair of x_P and X_T indicator, period of the year, and crop. For each period of the year, crop and pair of climatic indicators, among the variants of eq. 2, we first selected the model with lowest Akaike Information criterion (AIC). AIC is a statistical performance measure that accounts for



the model complexity, thus allowing a fair comparison of models differing in number of coefficients. For each
crop and for each set of explanatory indicators, among the models with lowest AIC relative to the different periods,
we picked the one with highest complexity as the final model. With this strategy, we had models of comparable
complexity for separate crops and separate indicators.

The third type of model focused on short-duration, but potentially damaging, conditions. We considered four
short-term indicators as explanatory variables. These indicators were selected from the pool of reviewed indicators
(2.2.1), due to their damaging nature and reflected complementary behaviors, two reflecting precipitation
characteristics (NDP1, NDP20), and two temperature characteristics (NDT25 and Frost). The fixed part of the
model was

$$
\begin{aligned}
Yield = \beta_0 + \beta_t t &+ \beta_{NDP1}(NDP1) + \beta_{NDT25}(NDT25) + \beta_{NDP20}(NDP20) + \beta_{Frost}(Frost) \\
&+ \beta_{NDP1-2}(NDP1)^2 + \beta_{NDT25-2}(NDT25)^2 + \beta_{NDP20-2}(NDP20)^2 \\
&+ \beta_{Frost-2}(Frost)^2
\end{aligned}
\qquad \text{eq. 3}
$$

where $\beta_{NDP1}$ and $\beta_{NDP20}$ represent the linear dependencies of yield to two precipitation indicators NDP1 and
NDP20 and $\beta_{NDT25}$ and $\beta_{Frost}$ to two temperature indicators NDT25 and Frost, respectively. $\beta_{NDP1-2}$, $\beta_{NDP20-2}$,
$\beta_{NDT25}$ and $\beta_{Frost}$ represent the quadratic dependencies of the crops to the same indicators. We introduced
quadratic dependences to allow for yield-maximizing intermediate conditions and high levels of damaging
conditions. We did not include any interaction terms, because these conditions do not likely compound. We did
not perform any model selection and simplification for this model.

We applied the first type of model, that based on DI, to all four periods, including pre- growing season, while
the second and third types of models were applied on three periods, i.e., the entire growing season, pre-flowering
and after flowering.

We fitted each model via the restricted maximum likelihood method using fitlme function in MatLab R2023a.
Based on AIC, we determined which climatic indicators, model structures and periods had the highest performance
(Fig. 3). The same approach was used to compare the different lengths of the pre- (main) growing periods (30, 60,
90 days). Beyond AIC, we considered the fraction of explained variance by fixed effects only (r2marg) or by both
fixed effects and random effects (r2cond) as additional metric of model performance (Nakagawa & Schielzeth
2013). We considered effects significant when $p < 0.05$. We then plotted crop yield as a function of best performing
set of climatic indicators and pre- (main) growing dryness index, relative to the intermediate year 1992 (Fig. 4-5).

## 3  Results

The best performing model had precipitation sum and temperature averages as explanatory variables for winter
wheat (Fig. 3a) and spring barley (Fig. 3b), and maximum length of the dry spell (CDD) and temperature average
for spring wheat (Fig. 3c) and oats (Fig. 3d). The fraction of explained variance by fixed effects ranged from 0.15
to 0.50 (Fig. 3), and from 0.76 to 0.85 when considering also the random effects (Table 2; Supplementary
information SI, Table S1-S4). The fraction of explained variance was lowest for oats and highest for winter wheat.

Conditions relative to the entire (main) growing season had the highest performance for all crops, i.e., the
lowest AIC. Models based on post-flowering conditions for winter wheat (Fig. 3a) and oats (Fig. 3d), pre-flowering
conditions for spring barley (Fig. 3b), and pre-growing dryness index for spring wheat (Fig. 3c) had the second





best performance. However, despite a higher AIC, these models had comparable fraction of explained variance to the models of the entire growing season, for a given set of indicators. The difference in marginal r2 was 2-10%

depending on the crop (Fig. 3).

Models based on the dryness index had intermediate performance (Fig. 3) for all periods and crops except spring wheat. The effects of dryness index, as a proxy of the legacy effects of the period before the beginning of the (main) growing season, were best captured when considering the conditions during 60 days prior to the sowing for spring wheat, and 90 days for the other crops (not shown).

Models based on either short-term indicators or the variability of precipitation and temperature had the lowest performances (Fig. 3).





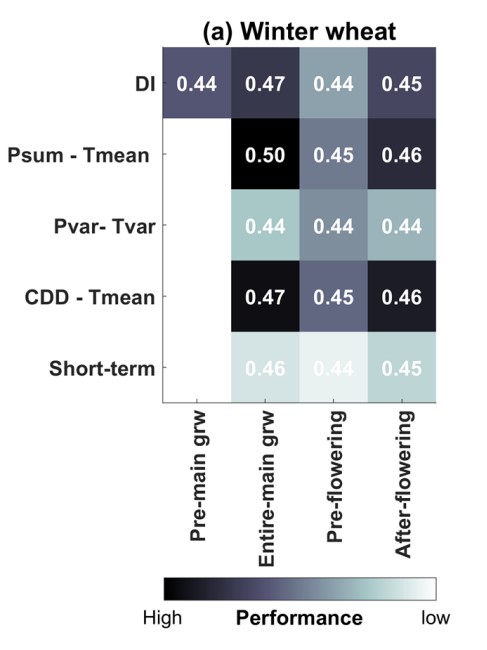

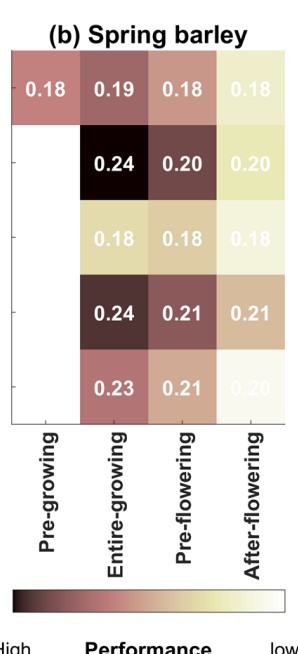

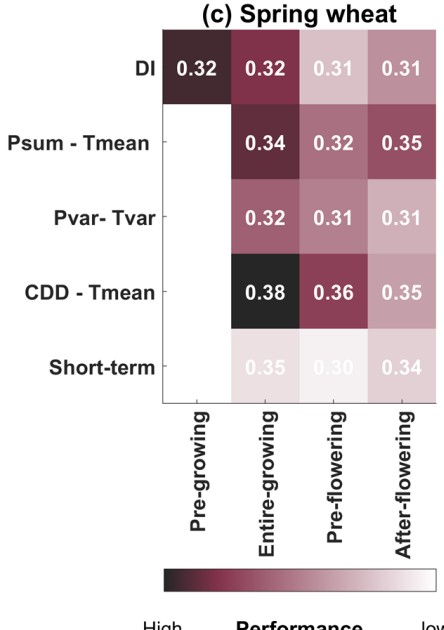

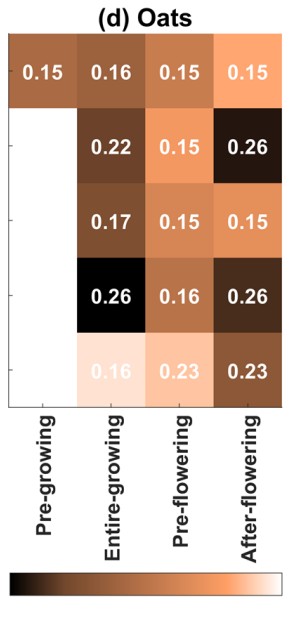

**Figure 3:** Performance of statistical models for yields differing in explanatory variables and period, for four crops (a) winter wheat (b) spring barley (c) spring wheat and (d) oats. Performance is assessed based on the AIC. The darker the color the lower the AIC





**and hence the higher the performance. Values in each cell indicate the corresponding r2marg for each model. Duration of the pre- (main) growing season is 2 months for spring wheat and 3 months for other crops.**

Yields increased between 0.2 ton·ha$^{-1}$ per decade for spring crops and 0.5 ton·ha$^{-1}$ per decade for winter wheat (Table 2), i.e., 7% to 10% of the long-term average.

**Table 2 Best performing combination of model, climatic indicators and period for each crop (a-d) and associated estimated coefficients, standard error (SE) and p. For each model we report also the fraction of explained variance when using only the fixed effects (r2marg) and when using both fixed effects and random effects (r2rand). Cases for p<0.05 are highlighted in bold.**

| | | Winter wheat and spring barley | | | | | |
|---|---|---|---|---|---|---|---|
| | Period | Entire growing season | | | | | |
| | Indicators | Psum (denoted as x_P) and Tmean (denoted as x_T) | | | | | |
| | Model structure | $\beta_0 + \beta_t t + \beta_P x\_P + \beta_T x\_T + \beta_{P2} x\_P^2 + \beta_{PT}(x\_P)(x\_T)$ | | | | | |
| | | a) Winter wheat | | | b) Spring barley | | |
| Predictor | Coefficient | Estimate | SE | p | Estimate | SE | p |
| - | $\beta_0$ [ton·ha$^{-1}$] | 10.432 | 1.253 | **<0.05** | 9.138 | 0.765 | **<0.05** |
| Time t | $\beta_t$ [ton·ha$^{-1}$·yr$^{-1}$] | 0.054 | 0.004 | **<0.05** | 0.028 | 0.002 | **<0.05** |
| Precipitation indicator P | $\beta_P$ [ton·ha$^{-1}$·cm$^{-1}$] | -0.330 | 0.074 | **<0.05** | -0.259 | 0.034 | **<0.05** |
| Temperature indicator T | $\beta_T$ [ton·ha$^{-1}$·°C$^{-1}$] | -0.568 | 0.088 | **<0.05** | -0.530 | 0.050 | **<0.05** |
| P x T | $\beta_{PT}$ [ton·ha$^{-1}$·cm$^{-1}$·°C$^{-1}$] | 0.032 | 0.005 | **<0.05** | 0.022 | 0.002 | **<0.05** |
| P$^2$ | $\beta_{P2}$ [ton·ha$^{-1}$· cm$^{-2}$] | -0.003 | 0.001 | **<0.05** | -0.001 | 0.000 | **<0.05** |
| | *r2marg* | 0.50 | | | 0.24 | | |
| | *r2cond* | 0.87 | | | 0.82 | | |
| | | Spring wheat and oats | | | | | |
| | Period | Entire growing season | | | | | |
| | Indicators | CDD (denoted as x_P) and Tmean (denoted as x_T) | | | | | |
| | Model structure | $\beta_0 + \beta_t t + \beta_P x\_P + \beta_T x\_T + \beta_{T2} x\_T^2 + \beta_{PT}(x\_P)(x\_T)$ | | | | | |
| | | c) Spring wheat | | | d) Oats | | |
| | Name | Estimate | SE | p | Estimate | SE | p |





| | | | | | | | |
|---|---|---|---|---|---|---|---|
| - | $\beta_0$ [ton·ha$^{-1}$] | -12.394 | 4.354 | **<0.05** | -9.544 | 2.905 | **<0.05** |
| Time t | $\beta_t$ [ton·ha$^{-1}$·yr$^{-1}$] | 0.033 | 0.003 | **<0.05** | 0.023 | 0.002 | **<0.05** |
| Precipitation indicator P | $\beta_P$ [ton·ha$^{-1}$·day$^{-1}$] | 0.130 | 0.054 | **<0.05** | 0.134 | 0.036 | **<0.05** |
| Temperature indicator T | $\beta_T$ [ton·ha$^{-1}$·°C$^{-1}$] | 2.116 | 0.612 | **<0.05** | 1.741 | 0.419 | **<0.05** |
| P x T | $\beta_{PT}$ [ton·ha$^{-1}$·day$^{-1}$·°C$^{-1}$] | -0.009 | 0.003 | **<0.05** | -0.011 | 0.002 | **<0.05** |
| T$^2$ | $\beta_{T2}$ [ton·ha$^{-1}$·°C$^{-2}$] | -0.071 | 0.021 | **<0.05** | -0.060 | 0.015 | **<0.05** |
| | *r2marg* | | 0.38 | | | 0.26 | |
| | *r2cond* | | 0.85 | | | 0.77 | |

In all crops, precipitation and temperature indicators interacted in defining yields (Table 2). Winter wheat and spring barley yields were maximum or near maximum for combinations of jointly increasing precipitation sums and average temperatures during the (main) growing season, with precipitation sum of 7 cm and average temperature of 12 °C for winter wheat, and precipitation sum of 11 cm and average temperature of 12 °C for spring barley (Fig. 4a-b). Yields of spring wheat and oats were maximum at approximately average temperature 14 °C

and one week-long CDD (Fig. 4c-d). The yield maximizing temperatures decreased with increasing CDDs.



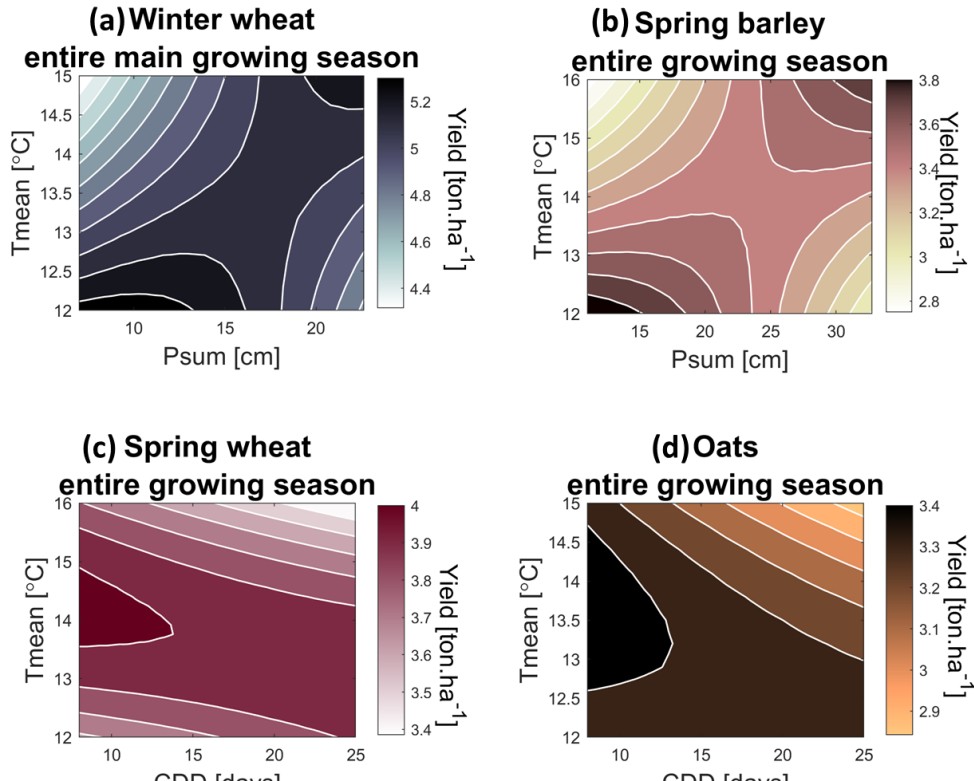

**Figure 4: Crop yield as a function of best performing set of climatic indicators. (a) winter wheat, (b) spring barley, (c) spring wheat (d) oats. Winter wheat and spring barley were best explained through precipitation sum (Psum) and temperature averages (Tmean) of the entire growing season, while spring wheat and pats were best explained through CDD and Tmean of the entire growing season. The contour plots are based on the fixed parts of the statistical models estimated for each crops separately. Time is set to the year 1992 which is an intermediate year within the study period (1965-2020). The ranges of the climatic indicators correspond to the 5th and 95th percentiles of each indicator.**

Yields of all crops except oats depended non-linearly on the pre-(main) growing dryness index (DI, Table 3). The yield maximizing DI was 1.2 mm·mm$^{-1}$ for winter wheat and at 2.7 mm·mm$^{-1}$ for spring wheat (Fig. 5a,c) – corresponding to the 60th and 90th percentiles of the observed range respectively. Also spring barley yield depended non-linearly on DI, but the yield maximizing DI was above the 95th percentile of the observed range of DI (Fig. 5b). For oats pre-growing proxy of water availability did not affect yield (Table 3d).

**Table 3 Model representing the legacy effects of water availability during pre-main growing period for winter wheat and pre-growing period for spring cereal. The table has the identified length of the period prior to (main) growing, model structure, estimated parameters and their units. Each model also included the fraction of explained variance when using only the field effects (r2marg) and when using both fixed effects and random effects (r2rand).**

| Crop | a) Winer wheat | b) Spring barley | c) Spring wheat | d) Oats |
|---|---|---|---|---|




| Identified length of the period prior to (main) growing | 90 days | | | 90 days | | | 60 days | | | 90 days | | |
|---|---|---|---|---|---|---|---|---|---|---|---|---|
| Model Structure | $\beta_0 + \beta_t t + \beta_{DI} DI_1 + \beta_{DI2} DI^2$ | | | | | | | | | | | |
| Name | Estimate | SE | p | Estimate | SE | p | Estimate | SE | p | Estimate | SE | p |
| $\beta_0$ [ton·ha⁻¹] | 3.147 | 0.285 | **<0.05** | 2.149 | 0.214 | **<0.05** | 2.747 | 0.228 | **<0.05** | 2.764 | 0.184 | **<0.05** |
| $\beta_t$ [ton·ha⁻¹·yr⁻¹] | 0.050 | 0.005 | **<0.05** | 0.025 | 0.003 | **<0.05** | 0.028 | 0.004 | **<0.05** | 0.021 | 0.004 | **<0.05** |
| $\beta_{DI}$ [ton·ha⁻¹] | 0.884 | 0.298 | **<0.05** | 0.195 | 0.055 | **<0.05** | 0.414 | 0.145 | **<0.05** | -0.053 | 0.065 | 0.41 |
| $\beta_{DI2}$ [ton·ha⁻¹] | -0.351 | 0.109 | **<0.05** | -.014 | 0.004 | **<0.05** | -0.075 | 0.032 | **<0.05** | 0.001 | 0.005 | 0.77 |
| *r2marg* | 0.44 | | | 0.18 | | | 0.32 | | | 0.15 | | |
| *r2cond* | 0.85 | | | 0.80 | | | 0.80 | | | 0.75 | | |

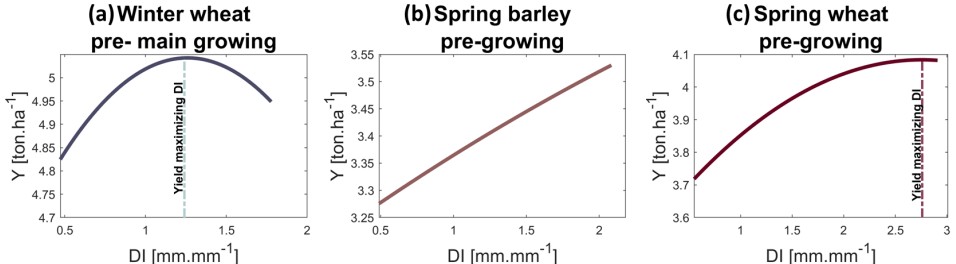

**Figure 5: Crop yield as a function of pre-growing dryness index (DI) based on the estimated length of legacy impacts for (a) winter wheat (3 months), (b) spring barley (3 months) and (c) spring wheat (2 months). The estimated yield is based on the fixed parts of the statistical models estimated for each crops separately. Time is set to 1992, the intermediate year in the study period 1965-2020. The ranges of the DIs correspond to the 5th and 95th percentiles of each indicator.**

## 4 Discussion

### 4.1 Average conditions explained crop yields better than short-term conditions

Models based on indicators representing average conditions performed better than those representing short-term, but potentially damaging, conditions (Fig. 3). This is in contrast with observed negative effects of heatwaves (i.e., temperatures exceeding 28 or 30 °C) at flowering in spring barley in Finland and winter barley in Minnesota (Hakala et al., 2020; Sadok et al., 2022). We surmise this low performance is in part due to the infrequent occurrence of short-term potentially damaging conditions (e.g., few occurrences of days with average temperatures above 25 °C, or frost during sensitive developmental stages). Moreover, these short-term conditions are easily averaged out across the cropped land in each county, either because of extreme-cancelling averaging of climatic conditions or because crops in different fields might not be simultaneously at the most sensitive stage.

Based on the AIC, models considering indicators relative to the entire (main) growing season had higher performance than those relative to sub-periods. Thus, crops were better explained by growing season conditions compared with those during a specific developmental stage. In accordance with our results, conditions during the entire growing season explained yield of spring cereals better than those of sub-periods in long-term experiments



in Sweden and Poland (Marini et al., 2020). This could be due to crops managing to compensate any unfavorable conditions during one period with increased growth during other periods (Foulkes et al., 2011). Nevertheless, over larger climatic gradients, the period around and after flowering explained yields better than the entire growing season (Hamed et al., 2022; Hoffman et al., 2020; Suliman et al., 2024), likely because these results refer to generally warmer conditions than in our study area, with temperatures at flowering likely exceeding the optimum. Moreover, crop development during the main growing season increases the relative importance of soil water evaporation and transpiration, and the dependence of the crop on water in the shallower layers, so that the same precipitation amount has different effects on crop growth. There are also uncertainties in estimating developmental stages through a model, such as that based on growing degree days used here, instead of observations, and by considering the model-estimated maturity date to coincide with the harvest date.

The explanatory power of climatic conditions were lower for oats and spring barley yields compared with spring and winter wheat (Fig. 4). A possible explanation is that wheat yield data refer to southern Sweden only, whereas spring barley and oats are grown under a wider range of latitudes and hence climatic conditions and other factors, such as soil conditions and day length. Moreover, the need to adapt to such a wide variety of conditions likely resulted in cultivation of different varieties, e.g. six- vs. two-row barley (Skoglund, 2022), which was not explicitly considered in our models.

Despite performing worse according to the AIC, conditions relative to sub-periods of the (main) growing season, as well as pre- (main) growing dryness index explained a fraction of yield variance comparable to that of the conditions during the entire (main) growing season for all crops. As such, conditions during sub-periods add complementary information to what can be deduced from the entire (main) growing season. Conditions before the main growing season can have practical relevance in terms of predicting water storage when precipitation is insufficient, and sowing date when the soil is water saturated, as well as an early prediction of low yields.

Dryness index during the period prior to the (main) growing season explained yields of all crops except spring barley. Dryness index during the 60-day period before the growing season was the second-best predictor of spring wheat yields, based on AIC, offering an early indication of how the final yield might be. The yields of other crops were better explained by dryness index averaged over 90 compared with 60 days. A potential explanation of the difference in period is that spring wheat is exclusively grown in southern Sweden (Fig. 1), where snowpack is shallower than in the north, reducing system memory in the form of accumulated snow. Early snowmelt and short snow-cover duration contributes to reduction of soil water in spring with sudden spikes in temperature (Pan et al., 2022), that can hamper crop establishment due to rapid drying of superficial soil layers.

## 4.2 Coordinated temperature and precipitation conditions maximized crop yields

In the best performing models, temperature averages interacted with precipitation sum when explaining winter wheat and spring barley yields, and with the length of the longest dry spells for spring wheat and oats yields. We surmise there is no physiological reason behind the different precipitation indicators, due to negligible differences between the fractions of explained variance and high ranking of both options in all crops. Rather, the response patterns to both precipitation sum and CDD point to the importance of sufficient water availability.

The temperature at which yields neared their maxima increased with precipitation sums and decreased with length of the dry spells (Fig. 4). These temperature by precipitation interactions are line with analyses of survey





data of wider climatic gradients (e.g., Carter et al., 2018; Luan et al., 2021; Matiu et al., 2017) and ecophysiological evidence that crop response to temperature depends on water availability (e.g., Suzuki et al., 2014) and hence precipitation and temperature. Both low precipitation and extended dry spells combined with high temperatures and, conversely, high precipitation sums under cool temperatures decreased yields, during the entire (main) growing season and pre-flowering period. In accordance with our results, yield declined with both excessive water

and dryer than normal conditions in Sweden (Sjulgård et al., 2023). Indeed, co-occurring high temperature and reduced water availability, but also cold and wet conditions, undermine crop growth (Ewel, 1999). High temperatures combined with long dry periods reduced yields through water and heat stress, by decreasing net photosynthesis and shortening the growth period (Fischer, 1985; Porter and Semenov, 2005; Slafer et al., 2023). However, yield losses due to high temperature averages were reduced by increasing precipitation (Peltonen-Sainio

et al., 2018; Schauberger et al., 2017; Tack et al., 2015), facilitating evaporative cooling associated with transpiration, particularly for oats and wheat (Martin et al., 2012; Schauberger et al., 2017). At the other extreme, wet and cool conditions was also damaging (Fig. 3) because such combination increases soil water content, can result in water logging, which in turn cause oxygen deficiency and reduced growth (Tian et al., 2021).

The composite indicator, dryness index (DI), inherently captures the interactive effects of temperature and

precipitation, by combining the potential evapotranspiration, which increasing with temperature, and precipitation (Luan et al., 2022). For the typical current pre-growing season DI (x-axis in Fig. 5) an increase in DI, i.e. an increase in temperature or a decrease in precipitation, was beneficial for spring crops, although absolute changes in yields were small. For winter wheat, increasing pre- main growing dryness up to intermediate DI of 1.2 mm·mm$^{-1}$ increased yield. Increasing DIs could be associated with reduced excessively wet soils that delay sowing and

decrease yields. Sowing time is determined by soil moisture content in shallow soil layers and field access to machinery (Trnka et al., 2011). For spring wheat that is grown in southern Sweden with little or no snow cover, a high DI is associated with early sowing and consequent high yields. Conversely, in Northern Sweden, high precipitation is associated with deep snow cover that takes longer time to melt in the spring, thus soil dries up slowly and sowing happens later. For winter wheat, excessive moist increases the risk of nitrogen loses through

denitrification and leaching (Guo et al., 2014). For all crops, removal of excessive moist enhances increasing soil temperatures that facilitates early crop growth and development (Porter and Semenov, 2005).

During pre-flowering, the range of required precipitation for winter wheat was lower than for spring wheat (SI, Fig. S1-S2). This is likely a consequence of winter wheat being already established prior to the start of the main growing season, i.e. with deeper roots that can benefit from the soil water that does not immediately dry up and

help the crop during early season dry periods (He et al., 2020; Thorup-Kristensen et al., 2020; Wang et al., 2017).

### 4.3 Hot or wet days were the only short-term conditions explaining yields

Among the considered short-term conditions (Table 1), only wet days and hot days (i.e., days with temperature above 25 °C) explained yields (SI, Table S1-S4). An intermediate number of wet days was associated to the maximum yields in all crops and periods, except for the pre-flowering period for spring wheat (SI, Table S2). This

is in line with physiological evidence and statistical analyses showing that increasing precipitation is beneficial only to a point, beyond which the disadvantages caused by water logging and reduced solar radiation dominate (Díaz-Torres et al., 2017). Hot days reduced yields in oats and spring barley (SI, Table S3-S4), in agreement with results from field experiments in Finland (Hakala et al., 2020). High temperatures during grain filling interfere



with starch synthesis and shorten the grain filling period (Sofield et al., 1977). While grain filling is quicker at
high temperatures, the final kernels become smaller and therefore high temperatures reduce yields (Hatfield &
Prueger, 2015).

## 4.4    Implications under climate change

Climatic indicators and time explained up to 85% of crop yield variability, in line with results from other
climates or extending globally (e.g., Luan et al., 2021; Ray et al., 2015; Zampieri et al., 2017). As such, the selected
indicators can provide insights on the impact of specific climatic conditions on cereals grown under Northern
European, including under future climates.

Total annual precipitation is projected to increase with climate change in Sweden (Grusson et al., 2021;
Teutschbein et al., 2023b), but precipitation increases are expected to occur during the winter, with no change or
even a slight decrease during summer (Breinl et al., 2020). The average annual temperature is expected to increase
by 2-6 °C by the end of the century, depending on greenhouse gas emission scenarios (IPCC, 2021). Considering
the projected warming, plant water availability is likely to decrease during the (main) growing season, when it is
most relevant for the crops.

In most locations globally, warming has negative effects on crops yields, as it accelerates development, leading
to shorter growth periods, and reduced accumulation of biomass before flowering and grains (Foulkes et al., 2011).
Although this pattern has been observed also at high latitudes e.g. in Finland (Peltonen-Sainio et al., 2011, 2015),
in the Nordic regions warmer temperatures are often expected to be beneficial by lengthening the growing season
(Slafer et al., 2023; Wiréhn, 2018). However, our results showed that benefits from higher temperatures were only
achieved when precipitation was higher or dry spells shorter. Conversely, warming reduced yields of all crops with
reduced or maintained average precipitation and maximum length of dry spells (Fig. 4). As such, the joint effects
of the expected changes in temperature and precipitation might be negative (Grusson et al., 2021). In line with our
results, yields were reduced by 50% on average in 2018 in Sweden (Beillouin et al., 2020; Statistiska-
Meddelanden, 2018), when summer temperature was 2.8 °C warmer than the 1981-2010 average temperatures and
CDD was 37 days, against a mean of 11.5 days in May August 1950-2020 (Teutschbein et al., 2023a; Wilcke et
al., 2020). Climate change is expected to make more frequent and intensify dry and hot periods, like the persistent
long and dry conditions that occurred in 2018 (included in our data), making the adverse climatic conditions the
new norm in the near future (Toreti et al., 2019).

Short-term intense precipitation can also become more frequent (Westra et al., 2014), causing mechanical
damage or increase excessive soil water (Iizumi et al., 2024). Abundant precipitation currently were associated to
reduced yields, although had low explanatory power (Fig. 3, SI, Tables S1-S4), partly due to infrequent occurrence
of extreme precipitation events during the study period.

Crops had their highest yields at temperature averages between 12 and 14 °C during the (main) growing season.
Cultivars bred and sown in Sweden have so far been selected to perform best under cool temperatures and short
growing seasons, compensated by long day lengths (Hakala et al., 2012). Warmer conditions might call for new,
better adapted varieties. At the same time, with increasing temperatures, optimal ranges of climatic conditions for
crop growth can be achieved beyond central-Sweden, opening new possibilities for wheat where it has been
historically difficult to cultivate it (Elsgaard et al., 2012). Increasing temperatures allows an earlier start of the





growing season, thus winter accumulated soil water can be used better and increase yields. However, higher summer temperatures counteract this positive effect by e.g. shortening the grain filling period (Sofield et al., 1977). Thus, the negative effect of increasing temperatures can to some extent be alleviated by shifting the growing season.

The timing of sowing of spring crop is and will remain critical under climate change. High soil water content in spring delays sowing of spring crops. On the contrary, soon after the optimal sowing time, soil moisture in superficial soil layers becomes too low to allow satisfactory germination and establishment of crops. There is a relatively small time-window that allows for sowing of spring crops between these two contrasting contributions of water. It is possible that future climates will not only shift but also further reduce such window of opportunity. Higher temperatures will cause faster soil water evaporation. This can advance sowing, by allowing earlier access to the field by the heavy machinery required for crop establishment. At the same time, low precipitation early in the growing season is common even for current climates in Sweden. Under climate change, insufficient precipitation can become more intense (Peltonen-Sainio et al., 2021), making establishment of spring crops difficult without irrigation.

A common adaptation to warmer and drier growing seasons is to replace spring cereals with winter wheat and other winter cereals. Winter cereals are already well established at the onset of the main growing season. They can, thus, handle precipitation deficits better than spring cereals, thanks to their more extensive root system that allow access to water deeper in the soil (Thorup-Kristensen et al., 2020). However, high precipitation in winter through climate change, combined with high temperatures will result in lower snow accumulation (Tootoonchi et al., 2023) that can negatively impact winter crops, due to the damaging effects of cold spells without the mitigating effects of snow (Vico et al., 2014). Milder winters can also facilitate damage from repeated freezing and thawing cycles or snow mold development (Andrews, 1996). As such, the net effects of altered winter and spring conditions on winter crops need further examination. An alternative adaptation measure to reduce failure risk from both insufficient and excessive water is mixing crops or varieties with different rooting depths and architectures allowing for the use of water at different depths (Manevska-Tasevska et al., 2024).

## 5    Conclusions

We systematically evaluated the role of various climatic indicators on explaining county-level cereal yields in Sweden for the period 1965-2020. Average growing season temperatures and precipitation totals, or maximum dry spell length explained yields best. Pre- and post- flowering, as well as pre- (main) growing season climatic conditions had a comparable range of explained variance, whereas short-term potentially damaging conditions, such as hot days and intense rainfall events, had the lowest explanatory power. High temperatures were associated with high yields, but only with adequate precipitation or sufficiently short dry spells. While we focused on crop yields in Sweden, our findings are likely applicable also to other high latitude regions.

Our results provide insights into the potential effects of climate change on crop yields at high latitudes. While warmer temperatures allow for wider areas to be cultivated by current crops, increasing temperature through climate change will not necessarily result in higher yields, if total precipitation during the main growing season does not increase. The potentially damaging short-term conditions were rare in the study period and did not explain yields, but they can become more frequent in the future. Adaptation to climate change will thus likely be necessary.



*Code and data availability:* Crop yield data is freely available from Statistics Sweden (Statistikdatabasen). Climatic data and land cover data are publicly available from E-Obs dataset (Cornes et al., 2018) and Corine land cover maps (European Environment Agency, 2020). All runs and analysis were performed with the Matlab coding language.

*Author contributions.* FT: Conceptualization, data curation, formal analysis, investigation, methodology, data
curation, visualization, software, writing and editing. GB: review and editing. GV: Conceptualization, methodology, supervision, funding acquisition, review and editing.

*Competing interests.* Authors declare no competing interests.

*Acknowledgments.* We acknowledge the partial support of the Swedish Research Council for Sustainable Development (FORMAS) (grant number 2023-02530). We also acknowledge the E-OBS dataset from the
Copernicus Climate Change Service (C3S, https://surfobs.climate.copernicus.eu) and the data providers in the ECA&D project (https://www.ecad.eu).

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
