# Peer review of "Warmer growing seasons improve cereal yields in Northern Europe only with increasing precipitation"

_EGUsphere, 2025_

## Community Comment (CC2)

I appreciate the authors response to my previous comment, however I do still have some remaining concerns. In the following, I have marked the authors reply to my comments in red (note I have only chosen to include parts of the authors reply that I still deem relevant for my concerns).

"The climatic indices we considered refer to the main growing season or the 30-60-90 days preceding that. Although annual climate averages differ substantially across Sweden, the temperature conditions relevant to crop development show less spatial variation. This is because the growing season occurs in partially different periods, depending on the location.

This is the case not only in Sweden, but also across larger latitudinal gradients: for example, the growing season average temperature from northern Sweden to southern Italy and Spain has been shown to be substantially aligned (Costa et al., 2024), despite the clear difference in annual average temperature. Lacking specific information on sowing date for spring crops and the release of dormancy for winter crops, we have used a Growing Degree Day (GDD)–based growing season. This approach explicitly adjusts the start and end of the main growing season according to the local temperature conditions. The GDD adjustment therefore normalizes much of the climatic contrast between north and south."

I agree that cropping-season temperatures can appear similar across distant latitudes, e.g., northern Sweden vs. southern Italy. This is because the comparable main cropping season occurs in fundamentally different parts of the year (winter–spring/early summer vs spring–summer). Within Sweden, however, the differences in cropping season are far less dramatic, and thus the analogy is not entirely appropriate. I also agree that, locally, agriculture is adapted to the particular seasonal window of the growing season. But what is important here is that the *length* of the growing season varies substantially within Sweden. See this figure below based on data from SMHI on the length of the growing season (1961–1990).

[Figure]

Besides at merely looking at seasonal temperatures or using a GDD-approach, we should also consider that in northern Sweden, the ground thaw occurs much later than in southern Sweden. Furthermore, the combination of rapidly reduced daylight hours and increased risk of early frosts in late summer and early autumn presents another definitive end to the actual growing season. These factors constitute *hard* limits to the growing season that are difficult to adapt to by merely adjusting the (nominal or real) sowing and harvesting dates. If this was not the case, grain production in northern Sweden would obviously be much larger.

Nonetheless, the main point of my comment was not that the climatic varies within Sweden, or that regions can experience different temperatures from year-to-year (temperatures of course correlates at quite large spatial scales at a seasonal resolution). Rather, it was that the response of harvest yields to temperatures, particularly in the summer, differs (where I would argue that the differences in the length of the *actual* growing season plays an important part).

The authors argue in their reply that:

"However, motivated by the comment received, we have now evaluated an alternative structure for the random factor, and specifically: Yield ~ ClimateVars + (Temperature ∣ County). This allows each region to have its own temperature–yield slope. This model including the random slopes did not improve model fit i.e., AIC did not decrease, and the estimated random slopes for temperature were very close to zero for most regions. This indicates that the effect of temperature on yield does not vary meaningfully across counties. In other words, the model itself provides evidence against the idea that climate–yield relationships are region specific, thus calling for separate models. A similar conclusion is reached if considering a random slope for the precipitation variable, i.e., Yield ~ ClimateVars + (Precipitation ∣ County)."

I want to underscore this part: **"This indicates that the effect of temperature on yield does not vary meaningfully across counties."** Taken at face value, this would imply that growing season temperatures exert the same influence on barley yields in Norrbotten, Västerbotten, and Jämtland as it does in Skåne, Gotland, and Blekinge. This is an extremely consequential conclusion, since it effectively contradicts over a century of empirical work using independent datasets spanning different periods.

Previous research, admittedly mainly using simple linear correlations and not regression models where other factors can be included and controlled for, have found that the yield response to temperature *do* meaning vary by county, as I mentioned in my first comment. Below I list five such studies, all covering different periods and using independent data.

1. **First, we have Axel Wallén (1917, 1918) who correlated barley yields from official statistics with instrumentally observed monthly summer temperatures during the period 1880–1910:**

[Figure]

Note the red color signifying negative correlations in response to mean July temperatures in large parts of southern Sweden, and dark black colors in the northernmost counties. For precipitation, we see the opposite pattern, albeit less significant.

2. **Edvinsson et al (2009) looked at subjective harvest assessments (of all the main grain crops combined) in the period 1723–1870:**

May–July precipitation correlates positively with harvest in southern Sweden (r = 0.58*), but not in northern Sweden (r = –0.21).

June–July temperature correlates negatively with harvest in the south (r = –0.37*) but positively in the north (r = 0.47*).

*significant at the p < 0.05 level.

3. **Skoglund (2022) (Harvest–climate relationships in Scania (Skåne), 1702–1911):**

Summer temperature correlates negatively with total grain tithes (r ≈ ‑0.23* to ‑0.26*).

Summer precipitation correlates positively (r ≈ 0.19* ‑ 0.36*).

Comparable patterns are seen for individual grains and in particularly so during the 1865–1911 period for spring cereals (see figure below).

[Figure]

**Figure 11.** Correlations of grain series vs. climate indicators 1865–1911. Note: Only statistically significant ($p \leq 0.05$) correlations are colored. Sources: SCB (2021), Seftigen et al. (2017, 2020), and SMHI (2021).

4. **Skoglund (2023) (Harvest–climate relationships in Jämtland, 1565–1911):**

Barley yields correlate strongly and positively with growing-season temperature ($r = 0.51$ and $0.61$ in two periods), and the effect persists in regression models controlling for annually variant sowing/harvest dates.

5. **Sjulgård et al. (2023) (Harvest–climate relationships 1965–2020 using apparently the same harvest data as the present study):**

Again supports a dipolar response pattern with clear regional/county differences.

[Figure]

**Fig. 2.** Pearson's correlation coefficients between yield anomalies of each crop group and Standardized Precipitation Evaporation Index (SPEI), heat wave index (HWI) and temperature anomalies (Temp) for each county based on crop yield and climate data from 1965 to 2020. The counties are sorted by decreasing latitude with the corresponding number from Fig. 1 and grouped into the northern, central or southern regions of Sweden. The brown colour shows a negative relationship to crop yield anomaly while blue colour represents a positive relationship. Non-significant (NS; $p > 0.05$) correlations are denoted by grey colour. White areas indicate counties with little or no cropping area of a certain crop group (NA). (For interpretation of the references to colour in this figure legend, the reader is referred to the web version of this article.)

In sum, since at least 1917, studies have got the same dipolar harvest yield response in relation to summer/growing season temperatures (at least for spring cereals), showing that the response *does* meaningfully differ between counties. This is why I find the authors conclusion to be so consequential, as it means all these previous studies, all using independent data and studying different periods (from the 16[th] century until the present day), are wrong. At the very least, I would hope that the authors can expand upon this and explain in more detail why all these previous studies need to be revised. Referring to AIC is not enough here, I think, as models can be specified and evaluated by many different means, AIC being just one popular, and for the most part robust, option. Alternatively, it could be that the models in the authors study are still not properly specified to account for these differences.

"We see that it is production, not productivity, having the ultimate implication in terms of locally produced cereal availability. However, we have chosen to focus on productivity, because we are primarily interested in the ecophysiological response of the different crops to a range of climatic conditions. In this way we can separate this effect from the combined effects of ecophysiological response and past and current spatial distribution of cultivated areas. We also note that the latter is less impacted by the year to year variation in climatic conditions compared with the productivity."

In your models, you are essentially giving the same weight to spring barley yields in Norrbotten (<0.01 % of the total in Sweden, based on averages 2000–2024), Västerbotten (0.02 % of the total in Sweden), and Skåne (30.42 % of the total) counties. This is, as you write, not a problem if you are just interesting the ecophysiological responses of different crops to climate. However, the problem arises in the interpretation of the results when you are taking the results from your models to draw conclusions for Sweden as a whole, and how future climate changes will affect yields. Obviously, there are huge differences between grain production between counties that needs to be taken into account in such an interpretation.

---

## Author Comment (AC1)

**Reply to reviewer #1 for on egusphere-2025-1982 "Warmer growing seasons improve cereal yields in Northern Europe only with increasing precipitation"**

*The reviewer's comments appear in black, our response in blue. Line numbers refer to the original submission.*

This manuscript systematically evaluated the effects of different climate factors (temperature, precipitation) on crop yield before and during the growing season, focusing on winter and spring cereal yields in Sweden from 1965 to 2020. Using county-level yield data and a range of physiologically relevant climate indicators for growth stages, the study found that warmer temperatures only benefit yields if accompanied by increased growing-season precipitation. This study, aligned with the focus of Biogeosciences on climate-ecosystem interactions, and provides recommendations for yield management in the context of climate warming in Sweden's high latitudes. However, the manuscript requires further improvement in its variable selection strategy, model specification, and reproducibility of the results. The article's structure and language should also be refined.

We thank the reviewer for their detailed comments that are helping us to further improve the manuscript. In a revised manuscript, we will clarify our methodology for feature selection, discuss and compare other modeling schemes. We have also identified ways to improve the discussion, by re-organizing and shortening it. The detailed responses to the specific comments can be found below.

Specific Comments:

1. Table 1: Nearly all the selected climate indicators represent frequency of short-term or extreme events, omitting intensity metrics that are known to influence crop yields. Please justify this choice or include intensity-based indicators in Section 2.2.1.

We fully agree that selected indicators must be complementary in nature. We selected a broad range of indicators reflecting both average conditions during whole growing season or subperiod, e.g. precipitation sum and temperature averages, other indicators reflecting frequencies of potentially damaging conditions, e.g. number of days with precipitation above 10 mm. In the selection we also considered the duration of these conditions, extending from short-term intense events to those with effects over longer periods. We also selected indicators that could have had effects over varying periods e.g. maximum number of consecutive dry days. While we acknowledge that even shorter-term damaging conditions, e.g. intense precipitation of 15 mins can have an adverse effect, this data is not readily available for the spatiotemporal scale of our study and is naturally concealed through daily averages. We will further clarify the rationale of the chosen indicators and any aspect unaccounted for in L130.

2. Model formulation:

- Why the Best formula can only include two main climate variables?

We considered together variables that are expected to have interactive effects on crops for physiological reasons (Luan et al., 2022; Ortiz-Bobea et al., 2021; Ray et al., 2015). While methodologically possible, combining more variables is not physiologically backed up and leads to complex models where interpretability is limited. Moreover, certain combinations, for example mixing wet days and dry days are not necessarily meaningful.

Nonetheless, we acknowledge that top-down feature selection approaches such as LASSO and random forest would allow for selections beyond two and free from priori assumptions. For

this reason, we checked selected features with random forest and will comment on the complementary nature of this method and the similar ones to our physiologically backed-up approach in section 2.2.1. We will consider whether to also add a sentence in the abstract to signal our bottom-up approach based on our physiological understanding.

- Pre-growing: why only consider DI, what about other indicators?

DI was chosen for the pre-growing period because this variable is the most likely one to capture effects that go beyond the inactive period of crop growth and extend into the main growing season for winter crops and the actual growing season for the spring crops. It is possible that also precipitation or extended dry periods alone could have an effect, as they also affect the amount of water available to the crop at least in the beginning of the main growing season. Average temperatures could also have an effect in defining the timing of snow melt and soil water evaporation and, in turn, soil water availability. However, precipitation and temperature before the main growing season are likely to affect crops together, by defining the soil water availability, making DI a superior choice. We will add a sentence to clarify this choice in L194.

- Rather than fitting pre- and post-flowering variables separately, perform LASSO selection on the full set of seasonal indicators simultaneously (as i*Zhu et al. The critical benefits of snowpack insulation and snowmelt for winter wheat productivity. Nat. Clim. Chang. 12, 485–490 (2022)*), then interpret the selected variables. Also you can try to put the average & short-term indicators together.

Thanks for this suggestion. We have chosen our candidate indicators and models as they strike a balance between feasibility, based on commonly available data, simplicity and interpretability in light of the eco-physiological understanding. In this way, we minimized the trial-and-error procedure of model selection and leaned toward a more process-oriented understanding.

However, we agree that our manuscript could benefit from further clarifications on our modeling approach. We are considering how to best compare our results with an automatic feature selection approach in section 2.2.1, bringing up the advantages/disadvantages with LASSO or random forest, also referring to Zhu et al. (2022) where we discuss the indicator representing the Sum of precipitation occurring when mean daily temperature is below 0 °C (P-T0), the proxy of snow amount, in L165.

- Many climate-crop studies use log-transformed yield as the dependent variable. Please clarify why raw yield Y was chosen. Will the result change if you use log(Y)?

This is typically done when the data is skewed and the linear mixed effect model assumptions are not satisfied. However, this was not the case for our data. Using untransformed yields has the further advantage that the fitted parameters are immediately interpretable, as changes in yields per unit change of conditions. We will clarify this in L204.

1. The potential role of snowfall on soil moisture and yield is also important in high latitudes regions. Please discuss or analyze snowfall indicators.

This is a very good point. While snowfall and ultimate snow-depth have a direct effect on winter crops and an indirect one on spring crops, looking at the snow would only explain the winter accumulation of moisture, because snow melts occur already by the end of March in the south and early May in north of Sweden. Furthermore, precipitation as rain nearer the main growing season, and whether that evaporates or not, matter as well. In other words, snow can contribute to soil water availability, but ultimately it is the early spring precipitation, combined with temperature, that defines water available early in the main growing season. Indeed winter crops could be directly affected by snow depth, but examining this direct effect would require

several snow- and winter-related indicators, adding additional indicators. Since we consider both spring and winter crops, we believe adding such indicators would make our work too complex and potentially confusing.

We also note that quantifying the direct effect of snow requires a reliable snow product. At the spatiotemporal scale of the current study, spanning from 1965 to 2020 and at 0.1° resolution, only a few gridded data sets are available, for which the uncertainties of the trends and the actual accumulation of snow is considerable (Wood et al., 2025). Relying on a proxy of snow depth (P-T0, i.e. the precipitation sum for days with temperature below 0 °C), only required precipitation and temperature data that are quality checked and more readily available for impact modelers, also over periods extending far in the past. Our correlation analyses showed that P-T0 is highly correlated with DI, and this contributed to the selection of DI for periods outside the main growing season, and extending over the winter. We will further clarify this choice and explain the possibility to use snow depth as an important indicator, in case of accessible reliable data in L165.

2. Section 2.2.2 Periods of interest: how to estimate the date of flowering and maturity. Now you only explain the beginning of growing-season. Uncertainty analyses of estimated phenological dates are also needed (e.g. shift the phenological dates forward or backward by equal intervals of days)

We have estimated the timing of flowering and maturity based on the growing degree day (GDD) approach, a common approach (e.g., Akyuz et al., 2017; Aslam et al., 2017; Liu et al., 2025), complementary to using crop growing calendar (e.g., Caparros-Santiago et al., 2021; Minoli et al., 2022). Through GDD approach, year-to-year variation of flowering and maturity dates caused by temperature are taken into account. This is particularly useful when field-level information regarding these phenological stages is not available. When revising the manuscript, we will adjust the description of the methods (L80-190), to clarify the approach used and its parameters (first day of counting, base temperature, thresholds for flowering and harvest).

We agree with the reviewer that such estimated dates are inherently uncertain. We thus checked for both the uncertainty caused by selecting different GDD thresholds and if there would be a difference in the results when year-to-year variations were considered instead of averages. None affected the results. We will clarify the lack of effects of the uncertainties in phenological stage estimation in L185.

3. Line300: here you indicated the yield maximizing DI, but for people who are not familiar with DI definition, they do not know what's the meaning of this DI value. What's the threshold of DI that represents drought? It should be clear in Methods.

That is a very good point and it is true that we missed explaining what DIs below or above 1 mean. We will clarify this in L165 and will remind the reader of the implications of DI in L300.

4. The current Discussion part repeats too many results and lacks logical flow. Please rewrite this part following a more clear structure and only remain main results.

Motivated by the reviewer's comment, we have carefully checked the discussion and found parts that can be removed or merged and reordered for improved flow and to avoid repetitions. More specifically, we have identified L356-370, L400-409, L465-470 to be revised. As we proceed with the revisions, it is likely that we will find additional parts that can be improved without loss of clarity.

Technical Corrections:

1. Figure 1b-c: The spatial map in panels b and c do not march the extent of panels a and d. The geographical boundaries should be the same as others.

We used the limited boundary to show that spring and winter wheat are only grown up to mid latitudes. We will add lines representing the Swedish borders for enhanced clarity.

2. Figure 2: the inclusion of too many indicators obscures the main information. Consider focusing on the most influential metrics. And the Spearman correlation coefficients were calculated from variables in which period? Please specify the period.

We see the reviewer's point that the number of indicators is high and the figure is complex. Nevertheless, we note that evaluating correlations is one of the steps we undertook towards reducing the number of indicators. For this reason, we would prefer to maintain all the indicators in this figure. However, we will use curly brackets to group them in three separate ones being precipitation, temperature and combination of precipitation and temperature. The figure refers to the entire growing season for winter wheat, estimated from sowing to maturity derived from the GDD model. We will clarify this in the caption.

3. Figure 3: add the explanation of each indicator in the caption (like what's short-term).

The caption will be revised as suggested

4. Table 2: relocate it to the Supplement and reference it appropriately in the main text.

We prefer to keep the table of coefficients (Table 2,3) in the main text. These are important parts of the results and guiding the reader on the significance of the parameters in the best fitting models. To keep the table brief, we had the indicators with lower ranking in the supplementary materials.

5. The current equations now only include fixed effects, random effects (e.g. site, year) also should be shown in the equation.

We will critically revise the text regarding the model, making sure the random effects are clearly indicated and justified. We prefer for the sake of clarity to write only the fixed part of the model. However, we see the reviewer's point that lacking an equation that includes also the random factors might be problematic. We will thus add the MatLab statements relative to the models to the supplementary materials. These clearly show the random factor structure and can also make our results reproducible.

6. Line275-280: "Yields increased between 0.2 ton·ha-1 per decade for spring crops and 0.5 ton·ha-1 per decade for winter wheat (Table 2), i.e., 7% to 10% of the long-term average". The location of this paragraph is a bit abrupt.

Upon re-reading the text, we agree. We will merge this sentence with L289-294.

**References**

Akyuz, F. A., Kandel, H., and Morlock, D.: Developing a growing degree day model for North Dakota and Northern Minnesota soybean, Agric. For. Meteorol., 239, 134–140, https://doi.org/10.1016/j.agrformet.2017.02.027, 2017.

Aslam, M. A., Ahmed, M., Stöckle, C. O., Higgins, S. S., Hassan, F. ul, and Hayat, R.: Can Growing Degree Days and Photoperiod Predict Spring Wheat Phenology?, Front. Environ. Sci., 5, https://doi.org/10.3389/fenvs.2017.00057, 2017.

Caparros-Santiago, J. A., Rodriguez-Galiano, V., and Dash, J.: Land surface phenology as indicator of global terrestrial ecosystem dynamics: A systematic review, ISPRS J. Photogramm. Remote Sens., 171, 330–347, https://doi.org/10.1016/j.isprsjprs.2020.11.019, 2021.

Liu, Z., Cammarano, D., Liu, X., Tian, Y., Zhu, Y., Cao, W., and Cao, Q.: Winter wheat yield responses to growing degree days: Long-term trends and adaptability in major producing areas of China, Ecol. Indic., 170, 113058, https://doi.org/10.1016/j.ecolind.2024.113058, 2025.

Luan, X., Bommarco, R., and Vico, G.: Coordinated evaporative demand and precipitation maximize rainfed maize and soybean crop yields in the USA, Ecohydrology, https://doi.org/10.1002/eco.2500, 2022.

Minoli, S., Jägermeyr, J., Asseng, S., Urfels, A., and Müller, C.: Global crop yields can be lifted by timely adaptation of growing periods to climate change, Nat. Commun., 13, 7079, https://doi.org/10.1038/s41467-022-34411-5, 2022.

Ortiz-Bobea, A., Ault, T. R., Carrillo, C. M., Chambers, R. G., and Lobell, D. B.: Anthropogenic climate change has slowed global agricultural productivity growth, Nat. Clim. Change, 11, 306–312, https://doi.org/10.1038/s41558-021-01000-1, 2021.

Ray, D. K., Gerber, J. S., MacDonald, G. K., and West, P. C.: Climate variation explains a third of global crop yield variability, Nat. Commun., 6, 5989, https://doi.org/10.1038/ncomms6989, 2015.

Wood, R. R., Janzing, J., van Hamel, A., Götte, J., Schumacher, D. L., and Brunner, M. I.: Comparison of high-resolution climate reanalysis datasets for hydro-climatic impact studies, Hydrol. Earth Syst. Sci., 29, 4153–4178, https://doi.org/10.5194/hess-29-4153-2025, 2025.

Zhu, P., Kim, T., Jin, Z., Lin, C., Wang, X., Ciais, P., Mueller, N. D., Aghakouchak, A., Huang, J., Mulla, D., and Makowski, D.: The critical benefits of snowpack insulation and snowmelt for winter wheat productivity, Nat. Clim. Change, 12, 485–490, https://doi.org/10.1038/s41558-022-01327-3, 2022.

---

## Author Comment (AC2)

**Reply to community comment by Martin Skoglund for on egusphere-2025-1982 "Warmer growing seasons improve cereal yields in Northern Europe only with increasing precipitation"**

*The reviewer's comments appear in black, our response in blue. Line numbers refer to the original submission.*

**General comments**

This study addresses a highly relevant and timely topic, and I find the authors' attempt to integrate short-, medium-, and long-term climatic influences on crop yields both commendable and innovative. The analysis represents a significant step forward compared to previous work by systematically evaluating multiple climatic indicators across different time-scales and their interactions across different crop types and periods.

We thank Martin Skoglund for their comments and suggestions.

However, I have several concerns regarding the model specification, which may substantially affect the results and their interpretation. These issues primarily concern how the models account for spatial heterogeneity in climatic responses across Sweden. In addition to this, I have some other smaller concerns regarding the interpretation of the results when using relative yield indicators as well as the formulations of the model equations.

1. **Model specification**

My main concern is that the models do not properly account for the well-known spatial differences in the temperature–precipitation dependency of crop yields between northern (N) and central/southern (C/S) Sweden. These regional contrasts were already formally identified by Wallén (1917, 1918) and have been confirmed in subsequent studies dealing with both historical and recent periods (e.g., Edvinsson et al. 2009; Skoglund 2022, 2023; Sjulgård et al. 2023).

In short:

• In northern Sweden (Norrland and Dalarna), higher summer temperatures are positively associated with yields, whereas increased precipitation often has a negative, albeit small, effect.

• In central/southern Sweden (counties south of Dalarna/Gävleborg), the relationship is generally the opposite: higher temperatures tend to reduce yields, while greater precipitation tends to increase them.

These contrasting relationships imply that aggregating N and C/S counties into a single model (this article) or time-series (see Holopainen et al. 2012), without explicitly accounting for these systematic differences, introduces biases and will underestimate the true climatic sensitivity of year-to-year yield variability.

This issue is especially relevant for spring-sown crops, such as barley, which are cultivated across the entire country, but also potentially for winter crops and wheat to a lesser extent (see Sjulgård et al. 2023). For winter crops and wheat, the shift from positive (negative) or negative (positive) relationships with temperature (precipitation) tend to occur only in the southernmost counties, if at all.

In the Discussion, the authors write:

"The explanatory power of climatic conditions were lower for oats and spring barley yields compared with spring and winter wheat … A possible explanation is that wheat yield data refer

to southern Sweden only, whereas spring barley and oats are grown under a wider range of latitudes and hence climatic conditions …"

Here the authors themselves imply that there is a possible aggregation bias, that lowers the explanatory power for oats and barley. The issue is less relevant for (spring and winter) wheat that is mainly grown in C./S. Sweden with a more homogenous climatic signal (see also the results for barley and wheat yields in Holopainen et al. 2012 where the same type of aggregation error is made). In an analysis where relationships are estimated at the county-level, where the systematic difference between N. and C./S. is largely accounted for (except perhaps when considering border counties such as Dalarna/Värmland/Gävleborg), it can clearly be seen that year-to-year climatic fluctuations have a much greater explanatory power in N. Sweden compared to C./S. Sweden (Sjulgård et al. 2023; Skoglund, 2022; 2023). In your model, this is mainly introduced as a random effect, which brings the oat and barley models to similar levels of explanatory power as the wheat models. However, because the random effects model assumes that group-level differences are uncorrelated with the explanatory variables, this treatment is inappropriate when the between-county differences are themselves driven by climate–yield dependencies.

Possible alternatives that address the aggregation bias:

- Including an interaction between region (N, C, and S or N and C/S) and key climatic variables.

- Fit separate models for N, C and S or N and C/S.

- A random-slope model that allows the effects of temperature and precipitation to vary by county.

We acknowledge that Sweden exhibits climatic differences across its parts. However, we disagree that the model must be split into separate subregions. We explain here our rationale.

We explicitly include aspects of the local climatic conditions, as summarized by the different climatic indices. In other words, the regional scale differences in climatic conditions are accounted directly, and not indirectly via a subdivision of the country. Of course, different parts of the country might experience different conditions, i.e., not being subject to all conditions represented in the contour plots.

The climatic indices we considered refer to the main growing season or the 30-60-90 days preceding that. Although annual climate averages differ substantially across Sweden, the temperature conditions relevant to crop development show less spatial variation. This is because the growing season occurs in partially different periods, depending on the location. This is the case not only in Sweden, but also across larger latitudinal gradients: for example, the growing season average temperature from northern Sweden to southern Italy and Spain has been shown to be substantially aligned (Costa et al., 2024), despite the clear difference in annual average temperature. Lacking specific information on sowing date for spring crops and the release of dormancy for winter crops, we have used a Growing Degree Day (GDD)–based growing season. This approach explicitly adjusts the start and end of the main growing season according to the local temperature conditions. The GDD adjustment therefore normalizes much of the climatic contrast between north and south.

County is included as random effect, i.e., we control for that and hence, implicitly, for the three regions mentioned above. We set county as random factor as we are interested in the crop response to the climatic conditions directly, not the location, which in turn affects the climatic conditions. We recognize that we set as random just the intercept though, i.e., we do not consider county-specific responses.

However, motivated by the comment received, we have now evaluated an alternative structure for the random factor, and specifically: Yield ~ ClimateVars + (Temperature | County). This allows each region to have its own temperature–yield slope. This model including the random slopes did not improve model fit i.e., AIC did not decrease, and the estimated random slopes for temperature were very close to zero for most regions. This indicates that the effect of temperature on yield does not vary meaningfully across counties. In other words, the model itself provides evidence against the idea that climate–yield relationships are region specific, thus calling for separate models. A similar conclusion is reached if considering a random slope for the precipitation variable, i.e., Yield ~ ClimateVars + (Precipitation | County).

Taken together, these points justify using single mixed-effects models across all cultivated regions of Sweden. The model effectively accounts for spatial and climatic heterogeneity while avoiding arbitrary regional divisions and maintaining statistical coherence. For these reasons, we will not change the model. However, we will add a sentence around section 2.3.3 (Linear mixed effect models) to better justify our modeling choice, while acknowledging the north-south climatic gradient and the ultimate yield differences, with reference to Sjulgård et al. (2023) and Skoglund (2022).

**Relevance of results:** Another issue, that is related to the model specification but only becomes an issue in regards to the interpretation of the results is that since you are treating yield as a relative variable (i.e., tonnes per hectare), instead of absolute production levels, the lumping together of N. and C./S. Sweden also obfuscates the social relevance of the results as the overwhelming majority of grain production occurs in C./S. Sweden.

We see that it is production, not productivity, having the ultimate implication in terms of locally produced cereal availability. However, we have chosen to focus on productivity, because we are primarily interested in the ecophysiological response of the different crops to a range of climatic conditions. In this way we can separate this effect from the combined effects of ecophysiological response and past and current spatial distribution of cultivated areas. We also note that the latter is less impacted by the year to year variation in climatic conditions compared with the productivity.

**Equations:** As a previous reviewer mentioned, in your model equations (eq. 1–3), you describe yield as, $Y$, when it should in fact be indexed as time- and location-variant, $Y_{it}$. Furthermore, all explanatory variables should also include $_{it}$ since they describe a given variable at time $_t$ and location $_i$. I do not believe the choice to make it only Y makes it clearer as the authors suggests as we now are several reviewers who had the same objection.

We will add the subscripts for clarity.

**References:**

Costa, A., Bommarco, R., Smith, M. E., Bowles, T., Gaudin, A. C. M., Watson, C. A., Alarcón, R., Berti, A., Blecharczyk, A., Calderon, F. J., Culman, S., Deen, W., Drury, C. F., Garcia y Garcia, A., García-Díaz, A., Hernández Plaza, E., Jonczyk, K., Jäck, O., Navarrete Martínez, L., Montemurro, F., Morari, F., Onofri, A., Osborne, S. L., Tenorio Pasamón, J. L., Sandström, B., Santín-Montanyá, I., Sawinska, Z., Schmer, M. R., Stalenga, J., Strock, J., Tei, F., Topp, C. F. E., Ventrella, D., Walker, R. L., and Vico, G.: Crop rotational diversity can mitigate climate-induced grain yield losses, Glob. Change Biol., 30, e17298, https://doi.org/10.1111/gcb.17298, 2024.

---

## Author Comment (AC3)

**Reply to community comment by Martin Skoglund for on egusphere-2025-1982 "Warmer growing seasons improve cereal yields in Northern Europe only with increasing precipitation"**

***The comments appear in black, as well as red when the commenter is quoting our previous response, our response in blue. Line numbers refer to the original submission.***

I appreciate the authors response to my previous comment, however I do still have some remaining concerns. In the following, I have marked the authors reply to my comments in red (note I have only chosen to include parts of the authors reply that I still deem relevant for my concerns).

"The climatic indices we considered refer to the main growing season or the 30-60-90 days preceding that. Although annual climate averages differ substantially across Sweden, the temperature conditions relevant to crop development show less spatial variation. This is because the growing season occurs in partially different periods, depending on the location.

This is the case not only in Sweden, but also across larger latitudinal gradients: for example, the growing season average temperature from northern Sweden to southern Italy and Spain has been shown to be substantially aligned (Costa et al., 2024), despite the clear difference in annual average temperature. Lacking specific information on sowing date for spring crops and the release of dormancy for winter crops, we have used a Growing Degree Day (GDD)–based growing season. This approach explicitly adjusts the start and end of the main growing season according to the local temperature conditions. The GDD adjustment therefore normalizes much of the climatic contrast between north and south."

I agree that cropping-season temperatures can appear similar across distant latitudes, e.g., northern Sweden vs. southern Italy. This is because the comparable main cropping season occurs in fundamentally different parts of the year (winter–spring/early summer vs spring–summer). Within Sweden, however, the differences in cropping season are far less dramatic, and thus the analogy is not entirely appropriate. I also agree that, locally, agriculture is adapted to the particular seasonal window of the growing season. But what is important here is that the *length* of the growing season varies substantially within Sweden. See this figure below based on data from SMHI on the length of the growing season (1961–1990).

[Figure]

The map provided in the comment refers to the vegetation period, which SMHI defines as the interval between the first day of a six-day period during which the mean daily temperature is at least +5.0 °C on all six days, and the last day before the first subsequent six-day period when the mean daily temperature falls below +5.0 °C on all six days, provided this occurs after 1 July (see https://www.smhi.se/en/climate/tools-and-inspiration/climate-indicators/length-of-the-vegetation-period , last accessed on Jan 13th, 2026). While this vegetation period encompasses the growing season of spring-sown crops and the main growing season of overwintering crops, it is generally longer than the actual growing season of annual field crops and is more relevant for perennial vegetation, such as trees, than for annual field crops. For example, in Kalmar County, in central eastern Sweden, spring cereals are typically sown in late April to early May and harvested from late July to mid-August, depending on the year, as captured by the growing degree day (GDD) model employed also in our work (see reply to Reviewer 2). This corresponds to a crop growing season of approximately 110–130 days, which is shorter than the vegetation period reported by SMHI for the same area. We disagree that the effective growing season of field crops is substantially longer in southern Sweden than in northern Sweden, despite the longer vegetation period in the south. With our data, the growing season has around 5 days difference between north (example Jämtland) and south (example Kalmar). This difference might be slightly bigger when considering the most favorable southernmost areas of Sweden, when crops can be sown also in late March or early April. However, the difference remains smaller than that apparent from the map reported above.

> Besides at merely looking at seasonal temperatures or using a GDD-approach, we should also consider that in northern Sweden, the ground thaw occurs much later than in southern Sweden. Furthermore, the combination of rapidly reduced daylight hours and increased risk of early frosts in late summer and early autumn presents another definitive end to the actual growing season. These factors constitute hard limits to the growing season that are difficult to adapt to by merely adjusting the (nominal or real) sowing and harvesting dates. If this was not the case, grain production in northern Sweden would obviously be much larger.

Indeed, these are valid constraints that explain why crop growing seasons do not span the entire vegetation period. Periods with temperatures above 5 °C for six consecutive days do not necessarily imply that soils are thawed, and crop growth therefore often begins later. Conversely, frost can occur before the average temperature falls below 5 °C for six consecutive days. In addition, harvest typically takes place in August or early September, well before the end of the vegetation period, because crops have reached physiological maturity rather than due to climatic limitations alone. Moreover, farmers select species and varieties that allow harvest in still dry and warm conditions, to avoid issues with machine driving over wet soils and high moisture content in grains. Consequently, although sowing dates differ considerably between northern and southern regions, harvest dates differ less. This is because, once sown, crops in the north develop more rapidly due to long days and rapidly rising temperatures. The main reason for lower yields in the north is that sowing takes place closer to summer when temperatures are high, leading to rapidly developing crops and less growth per development stage than further south where crops develop at lower temperatures. With more growth per development stage, there is less reduction of tillers and flower primordia, and therefore more fertile flowers per area are produced in the south than in the north. Conversely, even in the north, late growing season frost, if it occurs at all, is not a problem for yield formation, because frost generally comes when photosynthesis has already stopped and the crop is only maturing.

> Nonetheless, the main point of my comment was not that the climatic varies within Sweden, or that regions can experience different temperatures from year-to-year (temperatures of course correlates at quite large spatial scales at a seasonal resolution). Rather, it was that the response of harvest yields to temperatures, particularly in the summer, differs (where I would argue that the differences in the length of the actual growing season plays an important part).

> The authors argue in their reply that:

"However, motivated by the comment received, we have now evaluated an alternative structure for the random factor, and specifically: Yield ~ ClimateVars + (Temperature | County). This allows each region to have its own temperature–yield slope. This model including the random slopes did not improve model fit i.e., AIC did not decrease, and the estimated random slopes for temperature were very close to zero for most regions. This indicates that the effect of temperature on yield does not vary meaningfully across counties. In other words, the model itself provides evidence against the idea that climate–yield relationships are region specific, thus calling for separate models. A similar conclusion is reached if considering a random slope for the precipitation variable, i.e., Yield ~ ClimateVars + (Precipitation | County)."

I want to underscore this part: "This indicates that the effect of temperature on yield does not vary meaningfully across counties." Taken at face value, this would imply that growing season temperatures exert the same influence on barley yields in Norrbotten, Västerbotten, and Jämtland as it does in Skåne, Gotland, and Blekinge. This is an extremely consequential conclusion, since it effectively contradicts over a century of empirical work using independent datasets spanning different periods.

Previous research, admittedly mainly using simple linear correlations and not regression models where other factors can be included and controlled for, have found that the yield response to temperature do meaning vary by county, as I mentioned in my first comment. Below I list five such studies, all covering different periods and using independent data.

**1. First, we have Axel Wallén (1917, 1918) who correlated barley yields from official statistics with instrumentally observed monthly summer temperatures during the period 1880–1910:**

[Figure]

Note the red color signifying negative correlations in response to mean July temperatures in large parts of southern Sweden, and dark black colors in the northernmost counties. For precipitation, we see the opposite pattern, albeit less significant.

**2. Edvinsson et al (2009) looked at subjective harvest assessments (of all the main grain crops combined) in the period 1723–1870:**

May–July precipitation correlates positively with harvest in southern Sweden (r = 0.58*), but not in northern Sweden (r = –0.21).

June–July temperature correlates negatively with harvest in the south (r = –0.37*) but positively in the north (r = 0.47*).

*significant at the $p < 0.05$ level.

**3. Skoglund (2022) (Harvest–climate relationships in Scania (Skåne), 1702–1911):**

Summer temperature correlates negatively with total grain tithes (r ≈ –0.23* to –0.26*).

Summer precipitation correlates positively (r ≈ 0.19*–0.36*).

Comparable patterns are seen for individual grains and in particularly so during the 1865–1911 period for spring cereals (see figure below).

[Figure]

**Figure 11.** Correlations of grain series vs. climate indicators 1865–1911. Note: Only statistically significant ($p \leq 0.05$) correlations are colored. Sources: SCB (2021), Seftigen et al. (2017, 2020), and SMHI (2021).

**4. Skoglund (2023) (Harvest–climate relationships in Jämtland, 1565–1911):**

Barley yields correlate strongly and positively with growing-season temperature ($r = 0.51$ and $0.61$ in two periods), and the effect persists in regression models controlling for annually variant sowing/harvest dates.

**5. Sjulgård et al. (2023) (Harvest-climate relationships 1965-2020 using apparently the same harvest data as the present study):**

Again supports a dipolar response pattern with clear regional/county differences.

[Figure]

**Fig. 2.** Pearson's correlation coefficients between yield anomalies of each crop group and Standardized Precipitation Evaporation Index (SPEI), heat wave index (HWI) and temperature anomalies (Temp) for each county based on crop yield and climate data from 1965 to 2020. The counties are sorted by decreasing latitude with the corresponding number from Fig. 1 and grouped into the northern, central or southern regions of Sweden. The brown colour shows a negative relationship to crop yield anomaly while blue colour represents a positive relationship. Non-significant (NS; $p > 0.05$) correlations are denoted by grey colour. White areas indicate counties with little or no cropping area of a certain crop group (NA). (For interpretation of the references to colour in this figure legend, the reader is referred to the web version of this article.)

In sum, since at least 1917, studies have got the same dipolar harvest yield response in relation to summer/growing season temperatures (at least for spring cereals), showing that the response does meaningfully differ between counties. This is why I find the authors conclusion to be so consequential, as it means all these previous studies, all using independent data and studying different periods (from the 16th century until the present day), are wrong. At the very least, I would hope that the authors can expand upon this and explain in more detail why all these previous studies need to be revised.

We thank the commenter for providing such a wide range of previous results. We disagree that our results or conclusions contrast with those presented in points 1-5 above, as explained next:

In the examples above, the analyses are done separately in each region, or county, correlating yields with temperature. Beyond considering the small difference in growing season discussed above, and its role in defining the climatic conditions, in our models we include a precipitation by temperature interaction term, which makes the model able to capture differences in response to temperature at different precipitation levels, if those exist. In other words, we do not impose a role of county per se but let that emerge via the model fitting. Indeed, precipitation conditions corresponding to different counties lead to different responses to temperature, i.e., the T x P interaction is significant. Southern counties are warmer and crops are more prone to moisture limitations, whereas northern counties are cooler and less frequently water-limited. Our model results show that increases in temperature are detrimental in the (warmer) south when accompanied by low precipitation, but beneficial or neutral in the (cooler) north where moisture is not as limiting. In other words, our results are in line with the qualitative patterns reported above, without requiring a somewhat arbitrary division of the country in regions but rather capturing the outcomes of the likely ecophysiological mechanism at play - positive effects of warming where temperatures are lower and water is generally sufficient, negative effects under already warm conditions, where a further increase in temperature can lead to limitations due to water availability. Indeed, Wallén (1917, 1918) found negative temperature-yield correlations in southern Sweden and positive correlations in the north, results corroborated by those of Edvinsson et al. (2009) and Sjulgård et al (2023). Skoglund (2022) showed a historically positive effect of increasing precipitation and negative effect of temperature in the southern-most region of Sweden, Skania. Skoglund et al. (2023) report a positive effect of warming in Jämtland, which is generally a cooler county compared with e.g. Skania. The apparent regional differences in temperature sensitivity documented in earlier studies can be interpreted as different evaluations of a common response under contrasting climatic regimes. This provides a coherent explanation for why

allowing county-specific temperature slopes did not improve the model, as indicated by higher AIC values, while remaining fully consistent with the evidence cited above. These plots even further support our general conclusion when it comes to precipitation-temperature interactions. We will refer to the mentioned papers and add a sentence in **L420** re-interpreting our results in terms of regions, i.e., making where in the country warming has been beneficial.

> Referring to AIC is not enough here, I think, as models can be specified and evaluated by many different means, AIC being just one popular, and for the most part robust, option. Alternatively, it could be that the models in the authors study are still not properly specified to account for these differences.

As per our previous response, we have several physiological reasons to set the model as we have done. We use the AIC to further support our choice. We agree that AIC is just one possible approach to evaluate a statistical model, but, as noted in the comment, it is a rather a common approach and a robust one. We deem this suitable as a further support to our choices.

> "We see that it is production, not productivity, having the ultimate implication in terms of locally produced cereal availability. However, we have chosen to focus on productivity, because we are primarily interested in the ecophysiological response of the different crops to a range of climatic conditions. In this way we can separate this effect from the combined effects of ecophysiological response and past and current spatial distribution of cultivated areas. We also note that the latter is less impacted by the year to year variation in climatic conditions compared with the productivity."

> In your models, you are essentially giving the same weight to spring barley yields in Norrbotten (<0.01 % of the total in Sweden, based on averages 2000–2024), Västerbotten (0.02 % of the total in Sweden), and Skåne (30.42 % of the total) counties. This is, as you write, not a problem if you are just interesting the ecophysiological responses of different crops to climate. However, the problem arises in the interpretation of the results when you are taking the results from your models to draw conclusions for Sweden as a whole, and how future climate changes will affect yields. Obviously, there are huge differences between grain production between counties that needs to be taken into account in such an interpretation.

We agree that cultivated areas and total production vary considerably across Sweden. However, our analysis focuses specifically on crop responses to climatic conditions, i.e., productivity per unit area. Accordingly, our results are interpreted in terms of expected future productivity, rather than overall food availability or food security. Focusing on productivity allows us to disentangle ecophysiological responses, which are driven by the intrinsic biological limits of each crop and variety, from changes in cultivated area, which can fluctuate and may continue to do so under changing conditions.

---

## Author Comment (AC4)

**Reply to reviewer #2 for on egusphere-2025-1982 "Warmer growing seasons improve cereal yields in Northern Europe only with increasing precipitation"**

*The reviewer's comments appear in black, our response in blue. Line numbers refer to the original submission.*

This paper explores the meteorological drivers of observed variability in crop yields in Sweden, investigating statistical relationships between key weather / climate indices and the yields of the four most commonly grown crops in the country over 55 years (1965-2020). Given the common assertion that anthropogenic climate change is beneficial for crops in higher latitudes, the conclusion that the potential benefits of warming may be limited by lack of increased precipitation is a very important one. There are also a number of extremely useful insights in the manuscript regarding the relative explanatory power of different variables, so overall I am supportive of this work.

We thank the reviewer for their positive feedbacks and further comments that are helping us to improve the manuscript. As detailed below, in a revised manuscript, we will discuss the lengthening of the growing season, which however does not impact our conclusions. We will also mention the effect of increasing $CO_2$ concentrations.

I note that the manuscript has already received a review from a referee and also a community review, to which the authors have already responded. Here I will focus on two additional points that appear to have been overlooked, which I recommend are addressed in a revised manuscript.

1. As the authors note, climate change is expected to increase the length of the growing season, and indeed this has already happened. eg. in southern Sweden, the meteorologically-defined growing season length has been starting earlier by between 2 and 4 days per decade between 1950 and 2022 (Miś and Tomczyk, 2025 https://doi.org/10.1007/s00704-025-05382-6). So, over the period analysed in the current study, the growing season may have lengthened by over 20 days in some places. However, the authors conduct their analysis using an average growing season length over the study period. Is there a risk that the results may have been affected by including events or averaging periods outside of the growing season in the earlier part of the study period but excluding some within the growing season in the later part? This seems particularly pertinent in the context of the authors remark at lines 317-320: "We surmise this low performance is in part due to the infrequent occurrence of short-term potentially damaging conditions (e.g., few occurrences of days with average temperatures above 25 °C, or frost during sensitive developmental stages)". I would be reassured if the authors could demonstrate that their conclusions are not sensitive to the use of an average growing season length over a period when the length has changed by a non-trivial amount.

This is a valid point. In our dataset, the growing season length increases by approximately 10 days on average over the study period, the result of the advancement of the sowing date and a smaller advancement of the maturity date, as determined by the growing degree (GDD) model (**Figure 1**).

[Figure]

**Figure 1:** Example of time series of the day of the year (y-axis) corresponding to sowing (purple down-pointing triangles), flowering date (green stars) and maturity (pink triangles pointing upwards) for spring wheat in Kalmar County, in south-east of Sweden.

Motivated by the reviewer's concern, we tested the sensitivity of our results to the use of year and county-specific growing seasons, and hence length, instead of a fixed county-specific average, in determining the climatic indices. This had no meaningful effect on the number of extreme events identified or the average conditions or the resulting conclusions.

2. The authors do not mention the potential impact of rising atmospheric CO2 concentrations on photosynthesis, transpiration and yield - see, for example, Rezaie et al (2023 https://doi.org/10.1038/s43017-023-00491-0) for a recent review. If this is already having an influence then it will be included within the continuous variable representing the combined effects of time elapsed in since 1965 (as described in lines 197-200) so I don't think it will alter the findings regarding the relative importance of the different meteorological drivers during the period of observations. However, given the potential non-linear effects of CO2 effects in the future, especially in how they may affect crop responses to drought and high temperatures, the existence of these influences and their implications should be highlighted as an outstanding uncertainty in relation to the results here. Eg. it should be mentioned in line 199 alongside climate change and technological improvements, and also discussed in section 4.4 (Implications under climate change).

We agree with the reviewer that explicitly mentioning the role of atmospheric $CO_2$ concentration is helpful for a more complete description of conditions. However, as recognized by the reviewer, the role of the increase in $CO_2$ concentration cannot be disentangled by that of other trends, for example technological improvements. We will thus mention the physiological effects of $CO_2$ increase, referring to Rezaei et al. (2023) in **L70**, and explicitly mention that this is conflated with other changes in time in the coefficient of time in **L410** where we discussed implications under climate change. We note that all plotted results pertain to an intermediate time point (the year 1992, an intermediate year within the study period) and hence implicitly an intermediate atmospheric $CO_2$ concentration, among other changes occurred in the 55 years considered.